# Two-sided fairness in rankings via Lorenz dominance

**Virginie Do**[1,2], **Sam Corbett-Davies**[1], **Jamal Atif**[2], **Nicolas Usunier**[1]

[1]Facebook AI

[2]LAMSADE, Université PSL, Université Paris Dauphine, CNRS, France

virginiedo@fb.com, scd@fb.com, jamal.atif@dauphine.psl.eu, usunier@fb.com

## Abstract

We consider the problem of generating rankings that are fair towards both users and item producers in recommender systems. We address both usual recommendation (e.g., of music or movies) and reciprocal recommendation (e.g., dating). Following concepts of distributive justice in welfare economics, our notion of fairness aims at increasing the utility of the worse-off individuals, which we formalize using the criterion of *Lorenz efficiency*. It guarantees that rankings are Pareto efficient, and that they maximally redistribute utility from better-off to worse-off, at a given level of overall utility. We propose to generate rankings by maximizing concave welfare functions, and develop an efficient inference procedure based on the Frank-Wolfe algorithm. We prove that unlike existing approaches based on fairness constraints, our approach always produces fair rankings. Our experiments also show that it increases the utility of the worse-off at lower costs in terms of overall utility.

## 1 Introduction

Recommender systems have a growing impact on the information we see and on our life opportunities, as they help us browse news articles, find a new job, house, or people to connect with. While the objective of recommender systems is usually defined as maximizing the quality of recommendations from the user's perspective, the recommendations also have an impact on the recommended "items". News outlets rely on exposure to generate revenue, finding a job depends on which recruiter gets to see our resume, and the effectiveness of a dating application also depends on who we are recommended to—and if we are being recommended, then someone else is not. *Two-sided fairness in rankings* is the problem of generating personalized recommendations by fairly mediating between the interests of users and items. It involves a complex multidimensional trade-off. Fairness towards item producers requires boosting the exposure of small producers (e.g., to avoid winner-take-all effects and popularity biases [1]) at the expense of average user utility. Fairness towards users aims at increasing the utility of the least served users (e.g., so that least served users do not support the cost of item-side fairness), once again at the expense of average user utility. The goal of this paper is to provide an algorithmic framework to generate rankings that achieve a variety of these trade-offs, leaving the choice of a specific trade-off to the practitioner.

The leading approach to fairness in rankings is to maximize user utility under constraints of equal item exposure (or equal quality-weighted exposure) [54, 7] or equal user satisfaction [6]. When these constraints imply an unacceptable decrease in average user utility, so-called "trade-offs between utility and fairness" [65, 41] are obtained by relaxing the fairness constraints, leading to the optimization of a trade-off between average user utility and a measure of users' or items' inequality.

Thinking about fairness in terms of optimal utility/inequality trade-offs has, however, two fundamental limitations. First, the optimization of a utility/inequality trade-off is not necessarily Pareto-efficient from the point of view of users and items: it sometimes chooses solutions that decrease the utility of some individuals without making anybody else better off. We argue that reducing inequalities by decreasing the utility of the better-off is not desirable if it does not benefit anyone. The second

35th Conference on Neural Information Processing Systems (NeurIPS 2021).

limitation is that focusing on a single measure of inequality does not address the question of how inequality is reduced, and in particular, which fraction of the population benefits or bears the cost of reducing inequalities.

In this paper, we propose a new framework for two-sided fairness in rankings grounded in the analysis of generalized Lorenz curves of user and item utilities. Widely used to study efficiency and equity in cardinal welfare economics [53], these curves plot the cumulative utility obtained by fractions of the population ordered from the worst-off to the best-off. A curve that is always above another means that all fractions of the populations are better off. We define fair rankings as those with non-dominated generalized Lorenz curves for users and items. First, this definition guarantees that fair rankings are Pareto efficient. Second, examining the entirety of the generalized Lorenz curves provides a better understanding of which fractions of the population benefit from an intervention, and which ones have to pay for it. We present our general framework based on Lorenz dominance in usual recommendation settings (e.g., music or movie recommendation), and also show how extend it to *reciprocal recommendation* tasks such as dating applications or friends recommendation, where users are recommended to other users.

We present a new method for generating rankings based on the maximization of concave welfare functions of users' and items' utilities. The parameters of the welfare function control the relative weight of users and items, and how much focus is given to the worse-off fractions of users and items. We show that rankings generated by maximizing our welfare functions are fair for every value of the parameters. Our framework does not aim at defining what parameters are suitable in general — rather, the choice of a specific trade-off depends on the application.

From an algorithmic perspective, two-sided fairness is challenging because items' utilities depend on the rankings of all users, requiring global inference. Previous work on item-side fairness addressed this issue with heuristic methods without guarantees or control on the achievable trade-offs. We show how the Frank-Wolfe algorithm can be leveraged to make inference tractable, addressing both our welfare maximization approach and existing item-side fairness penalties.

We demonstrate that our welfare function approach enjoys stronger theoretical guarantees than existing methods. While it always generates rankings with non-dominated generalized Lorenz curves, many other approaches do not. We show that one of the main criteria of the literature, called equity of attention by Biega et al. [7], can lead to decrease user utility, while *increasing* inequalities of exposure between items. Moreover, equal user satisfaction criteria in reciprocal recommendation can lead to decrease the utility of *every user*, even the worse-off. Our notion of fairness prevents these undesirable behaviors. We report experimental results on music and friend recommendation tasks, where we analyze the trade-offs obtained by different methods by looking at different points of their Lorenz curves. Our welfare approach generates a wide variety of trade-offs, and is, in particular, more effective at improving the utility of worse-off users than the baselines.

We present our formal framework in Section 2. We discuss the theoretical properties of previous approaches in Section 3, and present our ranking algorithm in Section 4. Our experiments are described in Section 5, and the related work is discussed in Section 6.

## 2 Two-sided fairness via Lorenz dominance

### 2.1 Formal framework

**Terminology and notation.** We identify an item with its producer, so that "item utility" means "item producer's utility". The main paper focuses on fairness towards individual users and items. We describe in Appendix B the extension of our approach to sensitive groups of users or items. $|\mathcal{X}|$ denotes the cardinal of the set $\mathcal{X}$. Given $n \in \mathbb{N}$, we denote by $[\![n]\!] = \{1, \ldots, n\}$. The set of users $\mathcal{N}$ is identified with $\{1, ..., |\mathcal{N}|\}$ and the set of items $\mathcal{I}$ is identified with $\{|\mathcal{N}| + 1, ..., n\}$ where $n = |\mathcal{N}| + |\mathcal{I}|$. For $(i, j) \in \mathcal{N} \times \mathcal{I}$, we denote by $\mu_{ij}$ the value of item $j$ to user $i$.

A (deterministic) ranking $\sigma : \mathcal{I} \to [\![|\mathcal{I}|]\!]$ is a one-to-one mapping from items $j$ to their rank $\sigma(j)$. Following [54], we use *stochastic rankings* because they allow us to perform inference using convex optimization (see Section 4). The recommender system produces one stochastic ranking per user, represented by a 3-way *ranking tensor* $P$ where $P_{ijk}$ is the probability that $j$ is recommended to $i$ at rank $k$. We denote by $\mathcal{P}$ the set of ranking tensors.

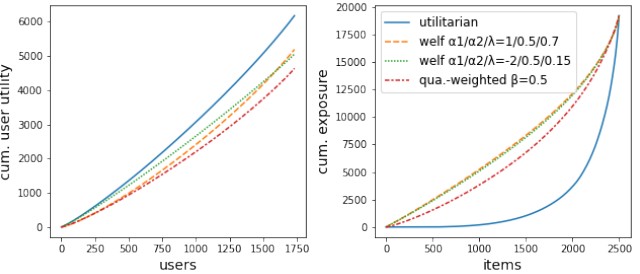
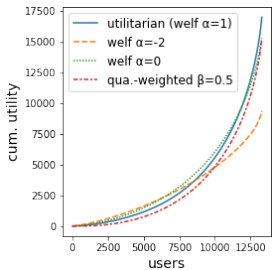

Figure 1: Generalized Lorenz curves for usual (left) and reciprocal (right) recommendation.

Utilities of users and items are defined through a position-based model, as in previous work [54, 7, 63]. Let $v \in \mathbb{R}^{|\mathcal{I}|}$, where $v_k$ is the exposure weight at rank $k$. We assume that lower ranks receive more exposure, so that $\forall k \in [\![|\mathcal{I}| - 1]\!], v_k \geq v_{k+1} \geq 0$.[1] Given a user $i$ and a ranking $\sigma_i$, the *user-side utility* of $i$ is the sum of the $\mu_{ij}$s weighted by the exposure weight of their rank $\sigma_i(j)$: $u_i(\sigma_i) = \sum_{j \in \mathcal{I}} v_{\sigma_i(j)} \mu_{ij}$. Given an item $j$, the *item-side utility* of $j$ is the sum over users $i$ of the exposure of $j$ to $i$. These definitions extend to stochastic rankings by taking the expectation over rankings, written in matrix form:[2]

$$\text{user-side utility:} \quad u_i(P) = \sum_{j \in \mathcal{I}} \mu_{ij} P_{ij} v \qquad \text{item-side utility (exposure):} \quad u_j(P) = \sum_{i \in \mathcal{N}} P_{ij} v$$

We denote by $\boldsymbol{u}(P) = (u_i(P))_{i=1}^n$ the utility profile for $P$, and by $\mathcal{U} = \{\boldsymbol{u}(P) : P \in \mathcal{P}\}$ the set of feasible profiles. For $\boldsymbol{u} \in \mathcal{U}$, $\boldsymbol{u}_\mathcal{N} = (u_i)_{i \in \mathcal{N}}$ and $\boldsymbol{u}_\mathcal{I} = (u_i)_{i \in \mathcal{I}}$ denote the utility profiles of users and items respectively.

**Two-sided fairness in rankings.** In practice, values of $\mu_{ij}$ are not known to the recommender system. Ranking algorithms use an estimate $\hat{\mu}$ of $\mu$ based on historical data. We address here the problem of *inference*: the task is to compute the ranking tensor given $\hat{\mu}$, with the goal of making fair trade-offs between (true) user and item utilities. Notice that the user-side utility depends only on the ranking of the user, but for every item, the exposure depends on the rankings of *all* users. Thus, accounting for both users' and items' utilities in the recommendations is a global inference problem.

**More general item utilities** We consider exposure as the item-side utility to follow prior work and for simplicity. Our framework and algorithm readily applies in a more general case of *two-sided preferences*, where items also have preferences over users (for instance, in hiring, job seekers have preferences over which recruiters they are recommended to). Denoting $\mu_{ji}$ the value of user $i$ to item $j$, the item side-utility is then $u_j(P) = \sum_{i \in \mathcal{N}} \mu_{ji} P_{ij} v$.

## 2.2 Lorenz efficiency and the welfare function approach

Our notion of fairness aims at improving the utility of the worse-off users and items. Since this does not prescribe exactly which fraction of the worse-off users/items should be prioritized, the assessment of trade-offs requires looking at all fractions of the population. This is captured by the generalized Lorenz curve used in cardinal welfare economics [53]. Formally, given a utility profile $\boldsymbol{u}$, let $(u_{(i)})_{i=1}^n$ be the sorted values in $\boldsymbol{u}$ from smallest to largest, i.e., $u_{(1)} \leq \ldots \leq u_{(n)}$, then the generalized Lorenz curve plots $(U_i)_{i=1}^n$ where $U_i = u_{(1)} + \ldots + u_{(i)}$. To assess the fairness of trade-offs, we rely on the following dominance relations on utility profiles:

**Pareto-dominance** $\succ_\text{P}$**.** $\boldsymbol{u} \succ_\text{P} \boldsymbol{u}' \iff (\forall i \in [\![n]\!], u_i \geq u'_i \text{ and } \exists i \in [\![n]\!], u_i > u'_i)$.

**Lorenz-dominance** $\succ_\text{L}$**.** Then $\boldsymbol{u} \succ_\text{L} \boldsymbol{u}' \iff \boldsymbol{U} \succ_\text{P} \boldsymbol{U}'$.

We write $\succeq_\text{L}$ for non-strict Lorenz dominance (i.e., $\forall i, U_i \geq U'_i$). Notice that Pareto-dominance implies Lorenz-dominance. Our notion of fairness, which we call *Lorenz efficiency*, states that a ranking is fair if the utility profiles for users and for items are not jointly Lorenz-dominated:

---

[1] We use a user-independent $v$ for simplicity. Considering user-dependent weights is straightforward.

[2] We consider $P_{ij}$ as a row vector in the formula, so that $P_{ij} v = \sum_{k=1}^{|\mathcal{I}|} P_{ijk} v_k$.

**Definition 1** (Lorenz efficiency). *A utility profile $\boldsymbol{u} \in \mathcal{U}$ is Lorenz-efficient if there is no $\boldsymbol{u}' \in \mathcal{U}$ such that either ($\boldsymbol{u}'_{\mathcal{I}} \succeq_{\mathrm{L}} \boldsymbol{u}_{\mathcal{I}}$ and $\boldsymbol{u}'_{\mathcal{N}} \succ_{\mathrm{L}} \boldsymbol{u}_{\mathcal{N}}$) or ($\boldsymbol{u}'_{\mathcal{N}} \succeq_{\mathrm{L}} \boldsymbol{u}_{\mathcal{N}}$ and $\boldsymbol{u}'_{\mathcal{I}} \succ_{\mathrm{L}} \boldsymbol{u}_{\mathcal{I}}$).*

We consider that Lorenz-dominated profiles are undesirable (and unfair) because the utility of worse-off fractions of the population could have been increased at no cost for total utility. Examples of Lorenz-curves of users and items are given in Fig. 1. The blue solid, green dotted and orange dashed curves are all non-dominated (the blue solid ranking has higher user utility but high item inequality, the green dotted and orange dashed curves have similar item exposure profiles, but user curves that intersect). On the other hand, the red dot/dashed curve is an unfair ranking: compared to the green dotted and orange dashed curve, all fractions of the worse off users have lower utility, together with less exposure for worse-off items.

A fundamental result from cardinal welfare economics is that concave welfare functions of utility profiles order profiles according to Lorenz dominance [3, 53]. The choice of the welfare function specifies which (fair) trade-off is desirable in a specific context. This result holds when all utilities are comparable. In our case where there are users and items, we propose the following welfare function parameterized by $\theta = (\lambda, \alpha_1, \alpha_2)$:[3]

$$\forall \boldsymbol{u} \in \mathbb{R}_+^n : \ W_\theta(\boldsymbol{u}) = (1-\lambda) \sum_{i \in \mathcal{N}} \psi(u_i, \alpha_1) + \lambda \sum_{j \in \mathcal{I}} \psi(u_j, \alpha_2) \ \text{ with } \psi(x, \alpha) = \begin{cases} x^\alpha & \text{if } \alpha > 0 \\ \log(x) & \text{if } \alpha = 0 \\ -x^\alpha & \text{if } \alpha < 0 \end{cases}.$$

Inference is carried out by maximizing $W_\theta$ (an efficient algorithm is proposed in Section 4):

$$\textit{(ranking procedure)} \qquad\qquad P^* \in \operatorname*{argmax}_{P \in \mathcal{P}} W_\theta(\boldsymbol{u}(P)) \qquad\qquad (1)$$

In $W_\theta$, $\lambda \in [0,1]$ controls the relative weight of users and items. The motivation for the specific choice of $\psi$ is that it appears in scale invariant welfare functions [43], but other families can be used as long as the functions are *increasing* and *concave*. Monotonicity implies that maxima of $W_\theta$ are Pareto-efficient. For $\alpha_1 < 1$ and $\alpha_2 < 1$, $W_\theta$ is strictly concave. Then, $W_\theta$ exhibits *diminishing returns*, which is the key to Lorenz efficiency: an increment in utility for a worse-off user/item increases welfare more than the same increment for a better-off user/item. The effect of the parameters is shown in Fig. 1 (left): For *item fairness* we obtain more item equality by using $\alpha_1 < 1$ (here, $\alpha_1 = 0.5$) and incrasing $\lambda$ (see blue solid vs orange dashed curve). The parameter $\alpha_2$ controls *user fairness*: smaller values yield more user utility for the worse-off users at the expense of total utility, with similar item exposure curve (green dotted vs orange dahsed curves). Let $\Theta = \{(\lambda, \alpha_1, \alpha_2) \in (0,1) \times (-\infty, 1)^2\}$. For every $\theta \in \Theta$, $W_\theta$ is strictly concave, and users and items have non-zero weight. We then have (the result is a straightforward consequence of diminishing returns, see Appendix C):

**Proposition 1.** $\forall \theta \in \Theta, \forall P^* \in \operatorname*{argmax}_{P \in \mathcal{P}} W_\theta(\boldsymbol{u}(P))$, $P^*$ *is Lorenz-efficient.*

**Relationship to inequality measures**  A well-known measure of inequality is the Gini index, defined as $1 - 2 \times \mathrm{AULC}$, where AULC is the area under the Lorenz curve. The difference between Lorenz and generalized Lorenz curves is that the former is normalized by the cumulative utility. This difference is fundamental: we can decrease inequalities while dragging everyone's utility to 0. However, this would lead to dominated *generalized* Lorenz curves. Interestingly, for *item-side* fairness, the cumulative exposure is a constant and thus trade-offs between user utility and item exposure inequality are not really problematic. However, for user-side fairness, the total utility is not constant and reducing inequalities might require dragging the utility of some users down for the benefit of no one.

**Additional theoretical results**  In App. C.2, we show that as $\alpha_1, \alpha_2 \to -\infty$, utility profiles tend to leximin-optimal solutions [43]. Leximin optimality corresponds to increasing the utility of the worst-off users/items one a a time, similarly to a lexical order. In App. C.3, we present an excess risk bound, which provides theoretical guarantees on the *true* welfare when computing rankings based on *estimated* preferences, depending on the quality of the estimates.

---

[3]$W_\theta(\boldsymbol{u}) = -\infty$ if $\alpha \leq 0$ and $\exists i, u_i = 0$. In practice, we use $\psi(x + \eta, \alpha)$ for $\eta > 0$ to avoid this case.

## 2.3 Extension to reciprocal recommendation

In reciprocal recommendation problems such as dating, the users are also items. The notion of fairness simplifies to increasing the utility of the worse-off users, which can in practice be done by boosting the exposure of worse-off users. Our framework above applies readily by taking $\mathcal{N} = \mathcal{I}$ and $n = |\mathcal{N}|$. The critical step however is to redefine the utility of a user to account for the fact that (1) the user utility comes from both the recommendation they receive and who they are recommended to, and (2) users have preferences over who they are recommended to.

To define this *two-sided utility*, let us denote by $\mu_{ij}$ the mutual preference value between $i$ and $j$, and our examples follow the common assumption that $\mu_{ij} = \mu_{ji}$ (see e.g., [45]). For instance, when recommending CVs to recruiters, $\mu_{ij}$ can be the probability of interview, while in dating, it can be that of a "match". The two-sided utility is then the sum of the user-side utility and item-sided utility of the user:

$$\overbrace{\overline{u}_i(P) = \sum_{j \in \mathcal{I}} \mu_{ij} P_{ij} v}^{\substack{\text{user-side utility} \\ \text{(}j\text{ recommended to }i\text{)}}} \qquad \overbrace{\overline{v}_i(P) = \sum_{j \in \mathcal{N}} \mu_{ij} P_{ji} v}^{\substack{\text{item-side utility} \\ \text{(}i\text{ recommended to }j\text{)}}} \qquad \overbrace{u_i(P) = \overline{u}_i(P) + \overline{v}_i(P)}^{\text{(two-sided) utility}}$$

With this definition of two-sided utility, our previous framework can be readily applied using $\mathcal{N} = \mathcal{I}$. A (two-sided) utility profile $\boldsymbol{u} \in \mathcal{U}$ is *Lorenz-efficient* if there is no $\boldsymbol{u}' \in \mathcal{U}$ such that $\boldsymbol{u}' \succ_{\mathrm{L}} \boldsymbol{u}$. The welfare function simplifies to $W_\theta(\boldsymbol{u}) = \sum_{i=1}^n \psi(u_i, \alpha)$, and Proposition 1 also holds true in this setting: maximizing the welfare function always yields Lorenz-efficient rankings.

Fig. 1 (right) illustrates how decreasing $\alpha$ increases utilities for the worse-off users at the expense of total utility. It also shows a Lorenz-dominated (unfair) profile, in which all fractions from the worst-off to the better-off users have lower utility.

From now on, we refer to *one-sided* recommendation for non-reciprocal recommendation.

## 3 Comparison to utility/inequality trade-off approaches

As stated in the introduction, leading approaches to fairness in ranking are based on utility/inequality trade-offs. We describe here the representative approaches we consider as baselines in our experiments. We then present theoretical results illustrating the undesirable behavior of some of them.

### 3.1 Objective functions

**One-sided recommendation**  In one-sided recommendation, the leading approach is to define exposure-based criteria for item fairness [54, 7]. The first criterion, *equality of exposure*, aims at equalizing exposure across items. The second one, *quality-weighted exposure*[4], which is advocated by many authors, defines the *quality* of an item as the sum of user values $q_j = \sum_{i \in \mathcal{N}} \mu_{ij}$ and aims for item exposure proportional to quality. The motivation of quality-weighted exposure is to take user utilities into account in the extreme case where the constraint is strictly enforced. Interestingly, as we show later, this approach has bad properties in terms of trading off user and item utilities.

In our experiments, we use the standard deviation as a measure of inequality. Denoting by $E = |\mathcal{N}| \, \|v\|_1$ the total exposure and by $Q = \sum_{j \in \mathcal{I}} q_j$ the total quality:

$$\substack{\textit{quality-weighted} \\ \textit{exposure}} \quad F_\beta^{\mathrm{qua}}(\boldsymbol{u}) = \sum_{i \in \mathcal{N}} u_i - \beta \sqrt{D^{qua}(\boldsymbol{u})} \ \text{ with } \ D^{qua}(\boldsymbol{u}) = \frac{1}{n} \sum_{j \in \mathcal{I}} \left( u_j - \frac{q_j E}{Q} \right)^2.$$

$$\substack{\textit{equality of} \\ \textit{exposure}} \quad F_\beta(\boldsymbol{u}) = \sum_{i \in \mathcal{N}} u_i - \beta \sqrt{D(\boldsymbol{u})} \quad \text{ with } \ D(\boldsymbol{u}) = \sum_{j \in \mathcal{I}} \frac{1}{n} \left( u_j - \frac{1}{|\mathcal{I}|} \sum_{j' \in \mathcal{I}} u_{j'} \right)^2.$$

Some authors use $D'(\boldsymbol{u}) = \sum_{(j,j') \in \mathcal{I}^2} |\frac{u_j}{q_j} - \frac{u_{j'}}{q_{j'}}|$ instead of $\sqrt{D^{qua}}$ [55, 42, 6]. $D^{qua}$ and $D'$ have qualitatively the same behavior. We propose $D^{qua}(\boldsymbol{u})$ as a computationally efficient alternative to $D'$, since it involves only a linear number of terms and $\sqrt{D^{qua}}$ is convex and differentiable except on $0$.

---

[4]We use here the terminology of [63]. This criterion has also been called "disparate treatment" [54], "merit-based fairness" [55] and "equity of attention" [7].

**Reciprocal recommendation** For reciprocal recommendation, we consider as competing approach a trade-off between total (two-sided) utility and inequality of utilities, as measured by the standard deviation:

*equality of utility*
$$F_\beta(\boldsymbol{u}) = \sum_{i \in \mathcal{N}} u_i - \beta \sqrt{D(\boldsymbol{u})} \quad \text{with } D(\boldsymbol{u}) = \sum_{j \in \mathcal{I}} \frac{1}{n} \left( u_j - \frac{1}{|\mathcal{I}|} \sum_{j' \in \mathcal{I}} u_{j'} \right)^2.$$

### 3.2 Inequity and inefficiency of some of the previous approaches

We point out here to two deficiencies of previous approaches.

First, for one-sided recommendation, we show that in some cases, compared to the welfare approach with any choice of the parameter $\theta \in \Theta$, quality-weighted exposure leads to the undesirable behavior of *decreasing user utility* while *increasing inequalities of exposure* between items. This is formalized by the proposition below, which uses the following notation: for $\theta \in \Theta$, let $\boldsymbol{u}^\theta = \mathrm{argmax}_{\boldsymbol{u} \in \mathcal{U}} W_\theta(\boldsymbol{u})$, and for $\beta > 0$, let $\mathcal{U}_\beta^{\mathrm{qua}} = \mathrm{argmax}_{\boldsymbol{u} \in \mathcal{U}} F_\beta^{\mathrm{qua}}(\boldsymbol{u})$.

**Proposition 2.** *The following claims hold irrespective of the choice of $\boldsymbol{u}^{\mathrm{qua},\beta} \in \mathcal{U}_\beta^{\mathrm{qua}}$.*

*For every $d \in \mathbb{N}_*$ and every $N \in \mathbb{N}_*$, there is a one-sided recommendation problem, with $d+1$ items and $N(d+1)$ users, such that $\forall \theta \in \Theta$, we have:*

$$\left( \exists \beta > 0, \boldsymbol{u}_\mathcal{N}^\theta \succ_{\mathrm{L}} \boldsymbol{u}_\mathcal{N}^{\mathrm{qua},\beta} \text{ and } \boldsymbol{u}_\mathcal{I}^\theta \succ_{\mathrm{L}} \boldsymbol{u}_\mathcal{I}^{\mathrm{qua},\beta} \right) \quad \text{and} \quad \lim_{\beta \to \infty} \frac{\sum_{i \in \mathcal{N}} u_i^{\mathrm{qua},\beta}}{\sum_{i \in \mathcal{N}} u_i^\theta} \xrightarrow[d \to \infty]{} \frac{5}{6}.$$

Second, in reciprocal recommendation, striving for pure equality can even lead to $0$ utility for *every user*, even that of the worst-off user. More precisely, we show that in some cases, compared to the welfare approach with any choice of parameter $\theta \in \Theta$, there exists $\beta > 0$ such that equality of utility has lower utility for every user, eventually leading to $0$ utility for everyone in the limit $\beta \to \infty$.

**Proposition 3.** *For $\beta > 0$, let $\mathcal{U}_\beta^{\mathrm{eq}} = \mathrm{argmax}_{\boldsymbol{u} \in \mathcal{U}} F_\beta(\boldsymbol{u})$. The claim below holds irrespective of the choice of $\boldsymbol{u}^{\mathrm{eq},\beta} \in \mathcal{U}_\beta^{\mathrm{eq}}$. Let $n \geq 5$. There is a reciprocal recommendation task with $n$ users such that:*

$$\forall \theta \in \Theta, \boldsymbol{u}^\theta, \exists \beta > 0 : \quad \forall i \in [\![n]\!], u_i^\theta > u_i^{\mathrm{eq},\beta} \quad \text{and} \quad \lim_{\beta \to \infty} \sum_{i \in \mathcal{N}} u_i^{\mathrm{eq},\beta} = 0.$$

**Proofs and additional results** All proofs are deferred to App. D, where we provide several additional results regarding the use of quality-weighted exposure and equality of exposure in reciprocal recommendation: We show in Prop. 8 that there are cases where both approaches lead to user utility profiles with Lorenz-dominated curves, and significantly lower total user utility than the welfare approach for any choice of the parameters.

## 4 Efficient inference of fair rankings with the Frank-Wolfe algorithm

We now present our inference algorithm for (1). Appendix E contains the proofs of this section and describes a similar approach for the objective functions of the previous section. From an abstract perspective, the goal is to find a maximum $P^*$ such that:

$$P^* \in \mathrm{argmax}_{P \in \mathcal{P}} W(P) \quad \text{with} \quad W(P) = \sum_{i=1}^n \Phi_i \left( \sum_{j=1}^n \mu_{ij}(P_{ij} + P_{ji})v \right)$$

where for every $i$, $\Phi_i : \mathbb{R}_+ \to \mathbb{R}$ is concave increasing, $\mu_{ij} \geq 0$ and $v$ is a vector of non-negative non-increasing values. Since $W$ is concave and $\mathcal{P}$ is defined by equality constraints, the problem above is a convex optimization problem. However, this is a global optimization problem over the rankings of all users, so a naive approach would require $|\mathcal{N}||\mathcal{I}|^2$ parameters and $2|\mathcal{N}||\mathcal{I}|$ linear constraints. The same problem arises with the penalties of previous work. In the literature, authors either considered applying the item-fairness constraints to each ranking individually [54, 6], which leads to inefficiencies with our definition of utility (see Appendix H), or resort to heuristics to compute the rankings one by one without guarantees on the trade-offs that are achieved [42, 7].

Our approach is based on the Frank-Wolfe algorithm [18], which was previously used in machine learning in e.g., structured output prediction or low-rank matrix completion [30], but to the best of

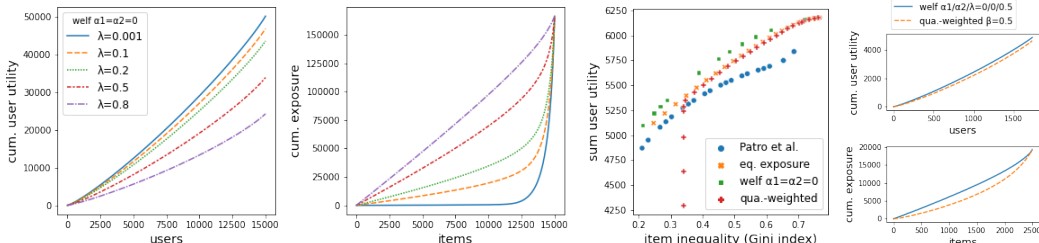

(a) Examples of generalized Lorenz curves achieved by *welf.* (b) Summary of trade-offs  (c) Dominated curve

Figure 2: Summary of results on Lastfm-2k, focusing on the user utility/item inequality trade-off.

our knowledge not for ranking. Denoting $\langle X \mid Y \rangle = \sum_{ijk} X_{ijk} Y_{ijk}$ the dot product between tensors, the algorithm creates iterates $P^{(t)}$ by first computing $\tilde{P} = \operatorname{argmax}_{P \in \mathcal{P}} \langle P \mid \nabla W(P^{(t)}) \rangle$ and then updating $P^{(t)} = (1 - \gamma^{(t)}) P^{(t-1)} + \gamma^{(t)} \tilde{P}$ with $\gamma^{(t)} = \frac{2}{t+2}$ [13]. Starting from an initial solution[5], the algorithm always stays in the feasible region without any additional projection step. Our main contribution of this section is to show that $\operatorname{argmax}_{P \in \mathcal{P}} \langle P \mid \nabla W(P^{(t)}) \rangle$ can be computed efficiently, requiring only one sort operation per user after computing the utilities. In the result below, for a ranking tensor $P$ and a user $i$, we denote by $\mathfrak{S}(P_i)$ the support of $P_i$ in ranking space.[6]

**Theorem 1.** *Let* $\tilde{\mu}_{ij} = \Phi_i'\big(u_i(P^{(t)})\big)\mu_{ij} + \Phi_j'\big(u_j(P^{(t)})\big)\mu_{ji}$. *Let* $\tilde{P}$ *such that:*

$$\forall i \in \mathcal{N}, \forall \tilde{\sigma}_i \in \mathfrak{S}(\tilde{P}_i): \quad \tilde{\sigma}_i(j) < \tilde{\sigma}_i(j') \implies \tilde{\mu}_{ij} \geq \tilde{\mu}_{ij'}. \text{ Then } \tilde{P} \in \operatorname*{argmax}_{\tilde{P} \in \mathcal{P}} \langle P \mid \nabla W(P^{(t)}) \rangle.$$

Moreover, it produces a compact representation of the stochastic ranking as a weighted sum of permutation matrices. The number of iterations of the algorithm allows to control the trade-off between memory requirements and accuracy of the solution. Using previous convergence results for the Frank-Wolfe algorithm [13], assuming each $\Phi_i''$ is bounded, we have:

**Proposition 4.** *Let* $B = \max_{i \in [\![n]\!]} \|\Phi_i''\|_\infty$ *and* $U = \max_{\boldsymbol{u} \in \mathcal{U}} \|\boldsymbol{u}\|_2^2$. *Let* $K$ *be the maximum index of a nonzero value in* $v$ *(or* $|\mathcal{I}|$*). Then* $\forall t \geq 1, W(P^{(t)}) \geq \max_{P \in \mathcal{P}} W(P) - O(\frac{BU}{t})$. *Moreover, for each user, an iteration costs* $O(|\mathcal{I}| \ln K)$ *operations and requires* $O(K)$ *additional bytes of storage.*

## 5 Experiments

### 5.1 One-sided recommendation

We first present experiments on movie recommendation task. We report here our experiments with the Lastfm-2k dataset [9, 47], which contains the music listening histories of $1.9k$ users. We present in App. F.2 experiments on a larger portion of the Last.fm dataset, and in App. F.3 results using the MovieLens-20m dataset [24]. Our results are qualitatively similar across the three datasets.

We select the top 2500 items most listened to, and estimate preferences with a matrix factorization algorithm using a random sample of $80\%$ of the data. All experiments are carried out with three repetitions for this subsample. The details of the experimental protocol are in App. F.1. Since the goal is to analyze the behavior of the ranking algorithms rather than the quality of the preference estimates, we consider the estimated preferences as ground truth when computing user utilities and comparing methods, following previous work. We compare our welfare approach (*welf*) to three baselines. The first one is the algorithm of [47] (referred to as *Patro et al.* in the figures), who consider envy-freeness for user-side fairness and, for item-side fairness, a constraint that the minimum exposure of an item is $\beta \frac{E}{|\mathcal{I}|}$ where $\beta$ is the trade-off parameter. The other baselines are quality-weighted exposure (*qua.-weighted*) and equality of exposure (*eq. exposure*) as described in Sec. 3.

---

[5]In our experiments, we initialize with the utilitarian ranking (Proposition 6).
[6]Formally, $\mathfrak{S}(P_i) = \big\{ \sigma : \mathcal{I} \to [\![|\mathcal{I}|]\!] \, \big| \, \sigma \text{ is one-to-one, and } \forall j \in \mathcal{I}, P_{ij\sigma(j)} > 0 \big\}$.

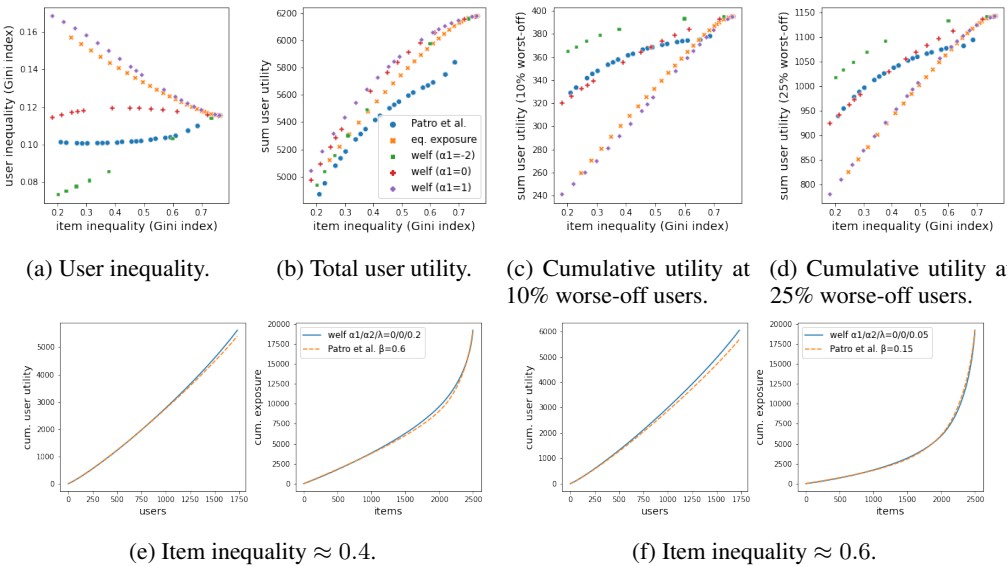

(a) User inequality.  (b) Total user utility.  (c) Cumulative utility at 10% worse-off users.  (d) Cumulative utility at 25% worse-off users.

(e) Item inequality $\approx 0.4$.  (f) Item inequality $\approx 0.6$.

Figure 3: Summary of results on Lastfm-2k for two-sided fairness: effect of varying $\alpha_1$.

**Item-side fairness** We first study in isolation item-side fairness, defined as improving the exposure of the worse-off item (producers). To summarize the trade-offs, we show the trade-offs by looking at exposure inequalities as measured by the Gini index (see Sec. 2.2). The results are given in Fig. 2:

- *Generating user utility/item inequality trade-offs* is performed with our approach by keeping $\alpha_1 = \alpha_2 = 0$ and varying the relative weight of items $\lambda$. Fig. 2a plots some trade-offs achieved by our approach. As expected, the user utility curve degrades as we increase the weight of items, while at the same time the curve of item exposure moves towards the straight line, which corresponds to strict equality of exposure. Fig. 6 in the appendix provides analogous curves for all methods, obtained by varying the weight $\beta$ of the inequality measure.
- *qua.-weighted yields unfair trade-offs* Fig. 2c shows a *welf* ranking that dominates a *qua.-weighted* ranking on both user and item curves. This is in line with the discussion of Section 3, *qua.-weighted* can lead to unfair rankings on utility/item inequality trade-offs.
- *welf dominates the user utility/item inequality (Gini) trade-offs* as seen on Fig. 2b: while all methods have the same total user utility when accepting high item inequality, *welf* dominates *Patro et al.*, *eq. exposure* and *qua.-weighted* as soon as Gini $\leq 0.5$. Note, however, that the Gini index is only one measure of inequality. When measuring item inequalities with the standard deviation, *eq. exposure* becomes optimal since our implementation optimizes a trade-off with this measure (see Fig. 8 in App. F.1). Overall, *welf* and *eq. exposure* yield different fair trade-offs.

**Two-sided fairness** Fig. 3 shows the effect of the user curvature $\alpha_1 \in \{-2, 0, 1\}$, keeping $\alpha_2 = 0$. Fig. 8 in App. F.1 shows similar plots when the item inequality is measured by the standard deviation rather than the Gini index.

- *Smaller $\alpha_1$ reduce user inequalities at the expense of total user utility, at various levels of item inequality.* This is observed by comparing the results for $\alpha_1 \in \{-2, 0, 1\}$ in Fig. 3a and Fig. 3b.
- *welf $\alpha_1 = 0$ is better than Patro et al.*, which can be seen by jointly looking at Fig. 3c, 3d and Fig. 3b which give the cumulative utility at different points of the Lorenz curve (10%, 25% and 100% of the users respectively). We observe that *welf $\alpha_1 = 0$* is similar to *Patro et al.* at the 10% and 25% levels, but has higher total utility. Example curves are given in Fig. 3e and 3f which plot *welf $\alpha_1 = 0$* and *Patro et al.* at two levels of item inequality. *welf $\alpha_1 = 0$* obtains similar curves to *Patro et al.*, except that it performs better at the end of the curve. A similar comparison can be made with *welf $\alpha_1 = 1$* and *eq. exposure*.
- *More user inequalities is not necessarily unfair* as seen in Fig. 3a comparing *welf $\alpha_1 = 0$* and *Patro et al.*. We observe that *welf $\alpha_1 = 0$* has slightly higher Gini index, but this is not unfair: as seen in Fig. 3e and 3f, this is due to the higher utility at the end of the generalized Lorenz curve of *welf*, but the worse-off users have similar utilities with *welf* and *Patro et al.*.

## 5.2 Reciprocal recommendation

We now present results on a reciprocal recommendation task, where fairness refers to increasing the utility of the worse-off users (this can be done by boosting their exposure at the expense of total utility). Since there is no standard benchmark for reciprocal recommendation, we generate an artificial task based on the Higgs Twitter dataset [15], which contains follower links, and address the task of finding mutual followers (i.e., "matches"). We keep users having at least 20 mutual links, resulting in a subset of 13k users. We build estimated match probabilities using matrix factorization. The experimental protocol is detailed in App. F.4. We also present in App. F.5 additional experiments using the Epinions dataset [49]. The results are qualitatively similar.

Our main baseline is equal utility (*eq. utility*) defined in Section 3. We also compare to quality-weighted exposure, and equality of exposure as baselines that ignore the reciprocal nature of the task. The results are summarized in Fig. 4:

- *Example of trade-offs obtained by varying $\alpha$* are plotted in Fig. 4a. As $\alpha$ decreases, the utility increases for the worse-off users at the expense of better-off users. We note that increasing the utility of worse-off users has a massive cost on total user utility: looking at the exact numbers we observe that $\alpha = -5$ has more than doubled the cumulative utility of the 10% worse off users compared to $\alpha = 1$ (120 vs 280), but at the cost of more than 60% of the total utility (17k vs 6.4k). Fig. 6 in Appendix F.4 contains plots of the trade-offs achieved by the other methods.
- *qua.-weighted and eq. exposure are dominated by welf on a large range of hyperparameters.* An example is given in Fig. 4b, where *welf* $\alpha = 0.5$ already dominates some of their models, even though in this region of $\alpha$ there is little focus on worse-off users. More generally, all values of $\beta \geq 0.1$ for *qua.-weighted* and *eq. exposure* lead to rankings with dominated curves. This is expected since they ignore the reciprocal nature of the task.
- *eq. utility is dominated by welf near strict equality* as illustrated in Fig. 4c: for large values of $\beta$, it is not possible to increase the utility of the worse off users, and *eq. utility* only drags utility of better-off users down.
- *welf is more effective at increasing utility of the worse-off users* as can be seen in Fig. 4e-g, which plots the total utility as a function of the cumulative utility at different points of the Lorenz curve (10%, 20%, 50% worse-off users respectively). For total utilities larger than 50% of the maximum achievable, *welf* significantly dominates *eq. utility* in terms of utility of worse-off users (10% and 25%) at a given level of total utility. *welf* also dominates *eq. utility* on the 50% worse-off users (Fig. 4h) in the interesting region where the total utility is within 20% of the maximum.
- *More inequality is not necessarily unfair* As shown in Fig. 4d, we see that for the same utility for the 10% worse-off users, *welf* models have higher inequalities than *eq. utility*. As seen before, this higher inequality is due to a higher total utility (and higher total utilities for the 25% worse-off users. The analysis of these Lorenz curves allow us to conclude that these larger inequalities are not due to unfairness. They arise because *welf* optimizes the utility of the worse-off users at lower cost in terms of average utility than *eq. utility*.

## 6 Related work

The question of fairness in rankings originated from independent audits on recommender systems or search engines, which showed that results could exhibit bias against relevant social groups [57, 33, 21, 40, 35] Our work follows the subsequent work on ranking algorithms that promote fairness of exposure for individual or sensitive groups of items [10, 8, 7, 54, 42, 65]. The goal is often to prevent winner-take-all effects, combat popularity bias [1] or promote smaller producers [39, 41]. Section 3 is devoted to the comparison with this type of approaches. Most of these works use a notion of fairness oriented towards items only. Towards two-sided fairness, Wang and Joachims [60] promote user-side fairness using concave functions of user utilities, similarly to us. Other works use equality constraints to define user-side fairness [6, 63]. These three approaches rely on the definitions of item-side fairness discussed in Section 3. Patro et al. [47] generate rankings that are envy-free on the user side, and guarantees the fair min-share for items. This approach is not amenable to controllable trade-offs between user and item utilities.

We are the first to address one-sided and reciprocal recommendation within the same framework. There is less existing work studying the fairness of rankings in the reciprocal setting. Xia et al. [64] aim at equalizing user utility between groups, which suffers from the problems discussed in

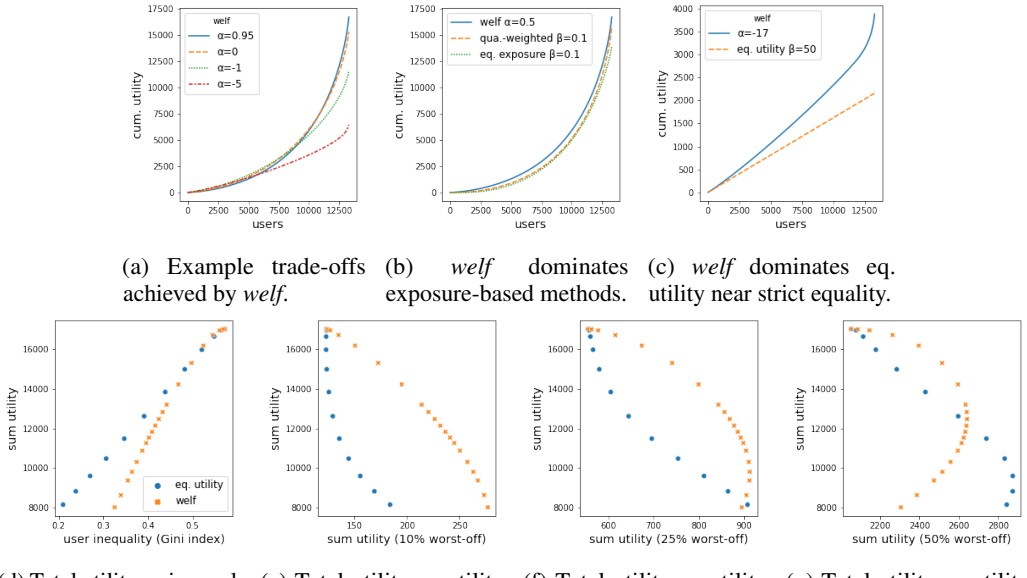

(a) Example trade-offs achieved by *welf*.

(b) *welf* dominates exposure-based methods.

(c) *welf* dominates eq. utility near strict equality.

(d) Total utility vs inequality.

(e) Total utility vs utility of 10% worse-off users.

(f) Total utility vs utility of 25% worse-off users.

(g) Total utility vs utility of 50% worse-off users.

Figure 4: Results on the twitter dataset.

Section 3, Jia et al. [31] generate rankings using a welfare function approach, but optimizing only the utility of users *being recommended*. Paraschakis and Nilsson [46] postprocess rankings to correct for inconsistencies between estimated and declared preferences of users. In contrast, we aim at fair trade-offs between user and item utilities, under the assumption that biases in the preference estimates have been addressed earlier in the recommendation pipeline. Fairness is also studied in the context of ridesharing applications [62, 37, 44], but they address matching rather than ranking problems.

There is growing interest in making the relationship between fairness in machine learning and social choice theory [25, 59, 4, 20, 27, 12, 16, 17], and welfare economics in particular [56, 28, 34, 36, 67]. In line with Hu and Chen [28], who focused on classification and parity penalties, we argue that Pareto-efficiency should be part of fairness assessments. We are the first to propose concave welfare functions and Lorenz dominance to address two-sided fairness in recommendation.

## 7 Conclusion

We view fairness in rankings as optimizing the distribution of user and item utilities, giving priority to the worse-off. Following this view, we defined fair rankings as having non-dominated generalized Lorenz curves of user and item utilities, and develop a new conceptual and algorithmic framework for fair ranking. The generality of the approach is showcased on several recommendation tasks, including reciprocal recommendation.

The expected positive societal impact of this work is to provide more principled approaches to mediating between several parties on a recommendation platform. Yet, we did not address several questions that are critical for the deployment of our approach. In particular, true user preferences are often not directly available, and we only observe proxies to them, such as clicks or likes. Second, interpersonal comparisons of utilities are critical in this work. It is thus necessary to make sure that the proxies we choose lead to meaningful comparisons of utilities between users. Third, estimating preferences or their proxies is itself not trivial in recommendation because of partial observability. The true fairness of our approach is bound to a careful analysis of (at least) these additional steps.

## Acknowledgments and Disclosure of Funding

We thank David Lopez-Paz, Jérôme Lang, as well as the anonymous NeurIPS reviewers for their constructive feedback on early versions of the paper.

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
