# A  Outline of the appendices

These appendices are structured as follows:

- In Appendix B, we present how our fairness framework can be applied to sensitive groups of users or categories of items.

- In Appendix C, we present a deeper analysis of the trade-offs achieved by the welfare approach. We also provide a theoretical guarantee relating the true welfare obtained by maximizing the welfare using estimated preferences, depending on the quality of the estimates.

- In Appendix D, we present the proofs for the theoretical results comparing our results and previous criteria of fairness in rankings. In addition, in Appendix D.3, we describe how to extend the criteria of equality of exposure and quality-weighted exposure in a reciprocal recommendation setting. This is the extension used in our experiments on reciprocal recommendation. In Proposition 8, we present an additional result regarding the inefficiency of these criteria in reciprocal recommendation.

- In Appendix E, we present the more general version of the Frank-Wolfe algorithm, which we use both to optimize the welfare function over stochastic rankings, as well as the penalty-based baselines. This appendix also contains the proofs of the results in Section 4. In addition, this appendix contains fundamental lemmas that are used in other appendices.

- Appendix F gives the details of the experiments presented in Section 5, as well we many additional experiments (two additional, larger scale datasets on one-sided recommendation, and an additional dataset for reciprocal recommendation)

- Appendix G briefly discusses the difference between the penalty we use in our implementation of the baseline approaches and an alternative penalty used by some authors.

- Finally, Appendix H discusses the difference between applying item-side fairness criteria for every ranking, compared to what we do in the paper, which defines item-side utility as an aggregate over the rankings of all users.

# B  Fairness towards sensitive groups rather than individuals

In all the paper we focus on fairness towards individual users and items rather than groups of users or items. Prior work [54, 42, 55] considered the utlity of a group as the sum or the average utility of its members. Using this definition of group utility, our framework dirrectly extends to groups rather than individuals. In this section we describe the case of one-sided recommendation with groups of users and item categories. The case of reciprocal recommendation (with user groups only) is similar but simpler.

Let $\mathcal{S} = (s_p)_{p=1}^{|\mathcal{S}|}$ be (possibly overlapping) user groups, i.e., $\forall p \in [\![|\mathcal{S}|]\!], s_p \subseteq \mathcal{N}$ and $\cup_{p \in [\![|\mathcal{S}|]\!]} s_p = \mathcal{N}$. Similarly, let $\mathcal{C} = (c_q)_{q=1}^{|\mathcal{C}|}$ be (possibly overlapping) item categories, i.e., $\forall q \in [\![|\mathcal{C}|]\!], c_q \subseteq \mathcal{I}$ and $\cup_{q \in [\![|\mathcal{C}|]\!]} c_q = \mathcal{I}$. On the user side, such groups would typically correspond to demographic groups considered sensitive for the application at hand [57]. On the item side, groups can represent a single producer for the case where we want to be fair to producers based on the aggregate utility they obtain from their products [41], or demographic groups as well [33].

In all cases, we redefine the user-side utility for groups and the item-side utility for categories:

$$u_{s_p}^{\mathrm{gr}}(P) = \sum_{i \in s_p} u_i(P) \qquad\qquad u_{c_q}^{\mathrm{cat}}(P) = \sum_{j \in c_q} u_j(P)$$

Let $\boldsymbol{u}^{\mathrm{gr}}(P) = (u_{s_p}^{\mathrm{gr}}(P))_{p=1}^{|\mathcal{S}|}$ and $\boldsymbol{u}^{\mathrm{cat}}(P) = (u_{c_q}^{\mathrm{cat}}(P))_{q=1}^{|\mathcal{C}|}$ be the utility profiles of user groups and item categories associated to $P$ respectively. The two-sided Lorenz efficiency for groups and categories is defined as:

**Definition 2.** *Let $\mathcal{S}$ be a set of user groups and $\mathcal{C}$ a set of item categories. Let $P \in \mathcal{P}$. $P$ is $(\mathcal{S}, \mathcal{C})$-Lorenz efficient if there is no $P' \in \mathcal{P}$ such that either condition holds:*

*1. $\boldsymbol{u}^{\mathrm{gr}}(P') \succeq_{\mathrm{L}} \boldsymbol{u}^{\mathrm{gr}}(P)$ and $\boldsymbol{u}^{\mathrm{cat}}(P') \succ_{\mathrm{L}} \boldsymbol{u}^{\mathrm{cat}}(P)$, or*

2. $\boldsymbol{u}^{\mathrm{cat}}(P') \succeq_{\mathrm{L}} \boldsymbol{u}^{\mathrm{cat}}(P)$ *and* $\boldsymbol{u}^{\mathrm{gr}}(P') \succ_{\mathrm{L}} \boldsymbol{u}^{\mathrm{gr}}(P)$.

The welfare function associated to $(\mathcal{S}, \mathcal{C})$, still parametrized by $\theta = (\lambda, \alpha_1, \alpha_2) \in \Theta$, is defined as

$$W_\theta^{\mathrm{gr}}(P) = (1 - \lambda) \sum_{s \in \mathcal{S}} \psi(u_s^{\mathrm{gr}}(P), \alpha_1) + \lambda \sum_{c \in \mathcal{C}} \psi(u_c^{\mathrm{cat}}(P), \alpha_2)$$

The welfare function follows the general form of objective function used for the algorithm in Appendix E, so the optimization of $W_\theta^{\mathrm{gr}}$ requires similar computational complexity as $W_\theta$.

Finally, the extension of Proposition 1 is straightforward. Its proof is similar to the proof presented in Appendix C.

**Proposition 5.** $\forall \theta \in \Theta, \forall P^* \in \underset{P \in \mathcal{P}}{\operatorname{argmax}} W_\theta^{\mathrm{gr}}(P)$, $P^*$ *is* $(\mathcal{S}, \mathcal{C})$-*Lorenz efficient.*

Note that this way of treating groups is not necessarily optimal. In particular, in does not account for within-group fairness. The separate consideration of within-group and between-group fairness has been studied extensively in the literature on equality of opportunity [50], which has inspired several works on algorithmic fairness [22, 26]. Yet, how to apply these principles to two-sided fairness in recommendation is still open, and is left as future work.

## C   More on welfare functions

This appendix provides an in-depth analysis of the trade-offs that are achievable by the welfare approach. We first pove the proposition of Section 2.2, and analyze the utilitarian rankings (obtained with $\alpha_1 = \alpha_2 = 1$). We then analyze how to obtain leximin optimal solutions on the side of the items in Appendix C.2, as mentioned in Section 2.2. Finally, we prove Theorem 2 in Appendix C.3, which provides a regret bound relating the true welfare achieved when maximizing welfare on estimated preferences. Some results in this section use Lemma 3 of Appendix 4, which is proved in Appendix 4.

Throughout the appendices, we use the more general version of item utilities (two-sided preferences), described at the end of Section 2.1. Moreover, to clarify the notation, we remind that a *ranking tensor* is a three-way tensor $P$ where $P_{ijk}$ is the probability that item $j$ is recommended to user $i$ at rank $k$. We consider $P$ as an $n \times n \times |\mathcal{I}|$ tensor, where irrelevant entries are set to 0. With this notation, the utility for both users and items can be written with the same formula:

$$\forall i \in [\![n]\!], u_i(P) = \sum_{j=1}^{n} \mu_{ij}(P_{ij} + P_{ji})v.$$

Note that this formula also corresponds to the two-sided utility in reciprocal recommendation. In general, the results in this appendix can be extended to reciprocal recommendation with minimal changes to their proofs, using $\mathcal{N} = \mathcal{I} = [\![n]\!]$ and the formula above for the utility.

### C.1   Lorenz efficiency and utilitarian ranking

We first prove Proposition 1:

**Proposition 1.** $\forall \theta \in \Theta, \forall P^* \in \underset{P \in \mathcal{P}}{\operatorname{argmax}} W_\theta(\boldsymbol{u}(P))$, $P^*$ *is Lorenz-efficient.*

*Proof.* It is well known that if $\Phi$ is increasing and strictly concave, then $F(\boldsymbol{u}) = \sum_{i=1}^{n} \Phi(u_i)$ is monotonic with respect to Lorenz dominance [53, 58]: $\boldsymbol{u} \succ_{\mathrm{L}} \boldsymbol{u}' \implies F(\boldsymbol{u}) > F(\boldsymbol{u}')$.

In the case of $W_\theta$, for every $\theta = (\lambda, \alpha_1, \alpha_2) \in \Theta$, both $\psi(., \alpha_1)$ and $\psi(., \alpha_2)$ are strictly concave by the definition of $\Theta$ (recall that in $\Theta$, we have $\alpha_1, \alpha_2 < 1$).

The partial function[7] $\boldsymbol{u}'_{\mathcal{N}} \mapsto W_\theta((\boldsymbol{u}_{\mathcal{I}}, \boldsymbol{u}'_{\mathcal{N}}))$ is, up to a constant, of the form of $F$ and likewise for the partial function $\boldsymbol{u}'_{\mathcal{I}} \mapsto W_\theta((\boldsymbol{u}'_{\mathcal{I}}, \boldsymbol{u}_{\mathcal{N}}))$. We now prove the result by contradiction. Assume that $\boldsymbol{u} \in \operatorname{argmax}_{\boldsymbol{u} \in \mathcal{U}} W_\theta(\boldsymbol{u})$ is not Lorenz efficient. Then there is $\boldsymbol{u}' \in \mathcal{U}$ such that $(\boldsymbol{u}'_{\mathcal{N}} \succeq_{\mathrm{L}} \boldsymbol{u}_{\mathcal{N}}$ and

---

[7]We denote by $(\boldsymbol{u}_{\mathcal{I}}, \boldsymbol{u}'_{\mathcal{N}})$ the vector $\mathbb{R}^d$ such that $(\boldsymbol{u}_{\mathcal{I}}, \boldsymbol{u}'_{\mathcal{N}})_i = u_i$ if $i \in \mathcal{I}$ and $(\boldsymbol{u}_{\mathcal{I}}, \boldsymbol{u}'_{\mathcal{N}})_i = u'_i$ if $i \in \mathcal{N}$.

$\boldsymbol{u}'_{\mathcal{I}} \succ_{\mathrm{L}} \boldsymbol{u}_{\mathcal{I}})$ or $(\boldsymbol{u}'_{\mathcal{N}} \succ_{\mathrm{L}} \boldsymbol{u}_{\mathcal{N}}$ and $\boldsymbol{u}'_{\mathcal{I}} \succeq_{\mathrm{L}} \boldsymbol{u}_{\mathcal{I}})$. Let us assume $(\boldsymbol{u}'_{\mathcal{N}} \succeq_{\mathrm{L}} \boldsymbol{u}_{\mathcal{N}}$ and $\boldsymbol{u}'_{\mathcal{I}} \succ_{\mathrm{L}} \boldsymbol{u}_{\mathcal{I}})$, the other case is dealt with similarly. We then have:

$$W_\theta(\boldsymbol{u}') \geq W_\theta((\boldsymbol{u}'_{\mathcal{I}}, \boldsymbol{u}_{\mathcal{N}})) \qquad\qquad \text{(because } \boldsymbol{u}'_{\mathcal{N}} \succeq_{\mathrm{L}} \boldsymbol{u}_{\mathcal{N}})$$
$$> W_\theta((\boldsymbol{u}_{\mathcal{I}}, \boldsymbol{u}_{\mathcal{N}})) \qquad\qquad \text{(because } \boldsymbol{u}'_{\mathcal{I}} \succ_{\mathrm{L}} \boldsymbol{u}_{\mathcal{I}})$$

which contradicts the maximality of $\boldsymbol{u}$. $\qquad\square$

The analogous for Proposition 1 for reciprocal recommendation is a direct consequence of standard results that concave welfare functions are monotonic with respect to Lorenz dominance [53, 58].

**Utilitarian ranking**  Proposition 6 below generalizes to two-sided utilities the well-known result that maximizing user-side utility is achieved by sorting $j \in \mathcal{I}$ by decreasing $\mu_{ij}$ (see e.g., [14]). For a ranking tensor $P$ and a user $i$, we denote by $\mathfrak{S}(P_i)$ the support of $P_i$ in ranking space.[8] We remind that $\sigma(j)$ is the rank of item $j$, and that lower ranks are better. For a user $i$ and item $j$, we use $\mu_{ji} = 1$.

**Proposition 6** (Utilitarian ranking). *Assume $\forall k \in [\![n-1]\!], v_k > v_{k+1} \geq 0$ and let*

$$P^* \in \operatorname*{argmax}_{P \in \mathcal{P}} W_{\frac{1}{2},1,1}(P) = \operatorname*{argmax}_{P \in \mathcal{P}} \sum_{i \in [\![n]\!]} u_i(P).$$

*1. $\forall i \in \mathcal{N}, \forall \sigma \in \mathfrak{S}(P_i^*) : \sigma(j) < \sigma(j') \implies \tilde{\mu}_{ij} \geq \tilde{\mu}_{ij'}$ with $\tilde{\mu}_{ij} = \mu_{ij} + \mu_{ji}$.*

*2. If $\forall (i,j) \in [\![n]\!]^2, \mu_{ij} = \mu_{ji}$, then $\tilde{\mu}_{ij} \geq \tilde{\mu}_{ij'} \iff \mu_{ij} \geq \mu_{ij'}$.*

When mutual preferences are symmetric (i.e., $\mu_{ij} = \mu_{ji}$), the utilitarian ranking is the same as the usual sort by decreasing $\mu_{ij}$. This also obviously holds when we consider exposue as item utility ($\mu_{ji} = 1$). This means that without considerations of two-sided fairness ($\alpha_1, \alpha_2 < 1$), the optimal ranking for two-sided utilities is the same as the usual ranking. This might explain why the two-sided utility has never been studied before, even in reciprocal recommendation [45].

For the proof of Proposition 6, the main part is the following lemma:

**Lemma 1.** *Let $F(\boldsymbol{u}(P)) = \sum_{i=1}^{n} u_i(P)$ and $\tilde{\mu}_{ij} = \mu_{ij} + \mu_{ji}$. Assume $\forall k \in [\![n-1]\!], v_k \geq v_{k+1} \geq 0$.*

*If $P^* \in \mathcal{P}$ is such that $\forall \sigma \in \mathfrak{S}(P_i^*), \forall j, j', \sigma(j) < \sigma(j') \implies \tilde{\mu}_{ij} \geq \tilde{\mu}_{ij'}$ then $P^* \in \operatorname*{argmax}_{P \in \mathcal{P}} \boldsymbol{u}(P)$.*

*Moreover, if $\forall k \in [\![n-1]\!], v_k > v_{k+1} \geq 0$, then the reciprocal is true.*

*Proof.* Notice that, thanks to the completion of $P$ with zeros on irrelevant entries and formula C, $F(\boldsymbol{u}(P))$ can be rewritten as:

$$F(\boldsymbol{u}(P)) = \sum_{i=1}^{n} u_i(P) = \sum_{i=1}^{n} \sum_{j=1}^{n} \mu_{ij}(P_{ij} + P_{ji})v = \sum_{i=1}^{n} \sum_{j=1}^{n} (\mu_{ij} + \mu_{ji})P_{ij}v$$

where the last equality is obtained by swapping $i$ and $j$ in the second sum, which is possible since $i$ and $j$ span the same range.

The result is then a direct consequence of Lemma 3 in Appendix E, using $A_{ij} = \mu_{ij} + \mu_{ji}$. $\qquad\square$

The first of statement of Proposition 6 assumes that the exposure weights $v$ are non-negative and strictly decreasing as per the second point of Lemma 1. Lemma 1 above gives the statement for the more general case of non-increasing $v$.

*Proof of Proposition 6.* The first statement is the consequence of Lemma 1 above, noticing that $F(\boldsymbol{u}(P))$ in Lemma 1 always has the same argmax. The second statement is obvious from the assumptions.

$\qquad\square$

---

[8]Formally, $\mathfrak{S}(P_i) = \big\{ \sigma : \mathcal{I} \to [\![|\mathcal{I}|]\!] \,\big|\, \sigma$ is one-to-one, and $\forall j \in \mathcal{I}, P_{ij\sigma(j)} > 0 \big\}$.

## C.2 Item-side leximin optimality

The most egalitarian trade-off achievable by our method is described by the leximin order [51]. Given two utility profiles $\boldsymbol{u}$ and $\boldsymbol{u}'$, $\boldsymbol{u} \geq_{\text{lex}} \boldsymbol{u}'$ if $\boldsymbol{U}$ is greater than $\boldsymbol{U}'$ according to the lexicographic order.[9] The leximin optimal profile is egalitarian in the sense that it maximizes the utility of individuals in sequence, from the worse-off to the better-off. Depending on the set of feasible profiles, this may not lead to equal utility for everyone, but any further reduction of inequality can only be achieved by making people worse off for the benefit of no other, in violation of Pareto-dominance.

The proposition below formalizes how leximin optimal solutions on the side of items are found. It shows that item-side leximin solutions are obtained by having $\alpha_2 \to -\infty$ and $\lambda \to 1$ at the same time. The proposition gives a formal statement of the rate at which $\lambda$ should converge to 1 relative to $\alpha$.

In the statement of the proposition, given two functions $F$ and $G$, we use $F(\alpha) \underset{\alpha \to -\infty}{\geq} G(\alpha)$ as a shorthand for $F(\alpha) \geq G(\alpha)$ for $\alpha$ sufficiently small.[10].

**Proposition 7.** *Let* $\mathcal{U}_{\text{lex}}^{\text{item}} = \{\boldsymbol{u} \in \mathcal{U} : \forall \boldsymbol{u}' \in \mathcal{U}, \boldsymbol{u}_{\mathcal{I}} \geq_{\text{lex}} \boldsymbol{u}'_{\mathcal{I}}\}$ *and let* $\boldsymbol{u}^* = \underset{\boldsymbol{u} \in \mathcal{U}_{\text{lex}}^{\text{item}}}{\operatorname{argmax}} \sum_{i \in \mathcal{N}} \psi(u_i, \alpha_1)$.

$$\forall \eta > \max(1, \|\boldsymbol{u}^*_{\mathcal{I}}\|_\infty), \forall \boldsymbol{u} \in \mathcal{U} : W_{1-\eta^\alpha, \alpha_1, \alpha}(\boldsymbol{u}^*) \underset{\alpha \to -\infty}{\geq} W_{1-\eta^\alpha, \alpha_1, \alpha}(\boldsymbol{u}).$$

This means that among the leximin-optimal item-side utility profiles, $\alpha_1$ still controls the redistribution profile on the user side, since it is possible that $|\mathcal{U}_{\text{lex}}^{\text{item}}| > 1$ in one-sided recommendation. A similar result holds for user-side item leximin.

*Proof.* Let $\boldsymbol{u}^* = \underset{\boldsymbol{u} \in \mathcal{U}_{\text{lex}}^{\text{item}}}{\operatorname{argmax}} \sum_{i \in \mathcal{N}} \psi(u_i, \alpha_1)$ and $\boldsymbol{u} \in \mathcal{U}$. Let $\theta = (\lambda, \alpha_1, \alpha)$ and take $\alpha < \min(0, \alpha_1)$.

Let $(j_1, j_2, \ldots, j_{|\mathcal{I}|})$ be the ranking of $\boldsymbol{u}^*_{\mathcal{I}}$ in increasing order: $u^*_{j_1} \leq \ldots u^*_{j_{|\mathcal{I}|}}$. Likewise, let $(j'_1, j'_2, \ldots, j'_{|\mathcal{I}|})$ be the ranking of $\boldsymbol{u}_{\mathcal{I}}$ in increasing order: $u_{j'_1} \leq \ldots \leq u_{j'_{|\mathcal{I}|}}$.

Let $m = \max\{k \in [\![|\mathcal{I}|]\!] \cup \{0\} : \forall \ell \leq k, u^*_{j_\ell} = u_{j'_\ell}\} + 1$, be the last index (+1) such that the smallest values of $\boldsymbol{u}^*$ and $\boldsymbol{u}$ are equal ($m = 1$ if the smallest values are different).

Let $C(\alpha) = W_{1-\eta^\alpha, \alpha_1, \alpha}(\boldsymbol{u}^*) - W_{1-\eta^\alpha, \alpha_1, \alpha}(\boldsymbol{u})$.

Let $K = \sum_{i \in \mathcal{N}} \left( \psi(u^*_i, \alpha_1) - \psi(u'_i, \alpha_1) \right)$.

**case 1:** $m = |\mathcal{I}| + 1$**.** Then $C(\alpha) = (1 - \eta^\alpha)K \geq 0$ since $\boldsymbol{u}^*_{\mathcal{I}} = \boldsymbol{u}_{\mathcal{I}}$ and $\boldsymbol{u}^*$ maximizes the user-side welfare.

**case 2:** $m < |\mathcal{I}|$**.** Then, we have $u_{j'_m} < u^*_{j_m}$ by the leximin optimality of $\boldsymbol{u}^*_{\mathcal{I}}$. We then have:

$$C(\alpha) = (1 - \eta^\alpha)K + \eta^\alpha \sum_{j \in \mathcal{I}} -(u^*_j)^\alpha + (u_j)^\alpha$$

$$= -(1 - \eta^\alpha)(u^*_{j_m})^\alpha \Big( \underbrace{\frac{K}{1 - \eta^\alpha} \Big( \frac{\eta}{u^*_{j_m}} \Big)^\alpha}_{\xrightarrow[\alpha \to -\infty]{} 0} + 1 + \sum_{k > m} \underbrace{\Big( \frac{u^*_{j_k}}{u^*_{j_m}} \Big)^\alpha}_{\xrightarrow[\alpha \to -\infty]{} 0} - \underbrace{\Big( \frac{u_{j'_m}}{u^*_{j_m}} \Big)^\alpha}_{\xrightarrow[\alpha \to -\infty]{} +\infty} - \sum_{k > m} \underbrace{\Big( \frac{u_{j'_m}}{u^*_{j_m}} \Big)^\alpha}_{\geq 0} \Big)$$

which implies $\lim_{\alpha \to -\infty} C(\alpha) = +\infty$ and thus the desired result. $\square$

## C.3 Guarantees when performing inference with estimated preferences

In practice, inference is carried out on an estimate $\hat{\mu}$ of $\mu$, meaning that, denoting $\hat{\boldsymbol{u}}$ the resulting estimated utility[11] the system output $\hat{P} = \operatorname{argmax}_{P \in \mathcal{P}} W_\theta(\hat{\boldsymbol{u}}(P))$. The following result extends

---

[9]Formally, $\boldsymbol{u} >_{\text{lex}} \boldsymbol{u}'$ if $(\exists k \in [\![d]\!]$ s.t. $\forall i < k, U_i = U'_i$ and $U_k > U'_k)$. $\boldsymbol{u} \geq_{\text{lex}} \boldsymbol{u}' \iff \neg(\boldsymbol{u}' \geq_{\text{lex}} \boldsymbol{u})$.

[10]Formally, $F(\alpha) \underset{\alpha \to -\infty}{\geq} G(\alpha) \iff \exists \alpha_0 \in \mathbb{R}, \forall \alpha \leq \alpha_0, F(\alpha) \geq G(\alpha)$.

[11]We have $\hat{\boldsymbol{u}}_i(P) = \sum_{j \in \mathcal{I}} \hat{\mu}_{ij} P_{ij} v$ for $i \in \mathcal{N}$.

surrogate regret bounds that exist in classification [5, 66] and learning to rank [14, 48, 2]to the case of welfare functions and global stochastic rankings. It makes the link between the quality of the estimate $\hat{\mu}$ and an optimality guarantee for $\boldsymbol{u}(\hat{P})$ (i.e., the true welfare of the ranking inferred on the estimated values). We prove the result for $\theta = (\frac{1}{2}, \alpha, \alpha)$ for $\alpha \leq 1$ to simplify notation.[12]

**Theorem 2.** *Let $\alpha \leq 1$ and $\theta = (\frac{1}{2}, \alpha, \alpha) \in \Theta$. Let $\hat{\mu} \in \mathbb{R}_+^{|\mathcal{N}| \times |\mathcal{I}|}$, $\hat{P} = \mathrm{argmax}_{P \in \mathcal{P}} W_\theta(\hat{\boldsymbol{u}}(P))$, and $P^* = \mathrm{argmax}_{P \in \mathcal{P}} W_\theta(\boldsymbol{u}(P))$.*

*Let furthermore $B(\hat{\mu}) = \max \big( \max_{i \in [\![n]\!]} \psi'(u_i(\hat{P}), \alpha), \max_{i \in [\![n]\!]} \psi'(\hat{u}_i(P^*), \alpha) \big)$. We have:*

$$W_\theta(\boldsymbol{u}(P^*)) - W_\theta(\boldsymbol{u}(\hat{P})) \leq 4B(\hat{\mu}) \sqrt{n \|v\|_2^2} \sqrt{\sum_{(i,j) \in \mathcal{N} \times \mathcal{I}} (\hat{\mu}_{ij} - \mu_{ij})^2}.$$

The existing results closest to our Theorem 2 are Theorem 2 of [14]. Here the result is substantially more difficult to prove because of the concave function and the fact that utilities are two-sided, calling for considering the rankings of multiple users at once.

*Proof.* We have:

$$W_\theta(\boldsymbol{u}(P^*)) - W_\theta(\boldsymbol{u}(\hat{P})) = W_\theta(\boldsymbol{u}(P^*)) - \underbrace{W_\theta(\hat{\boldsymbol{u}}(\hat{P}))}_{\geq W_\theta(\hat{\boldsymbol{u}}(P^*))} + W_\theta(\hat{\boldsymbol{u}}(\hat{P})) - W_\theta(\boldsymbol{u}(\hat{P}))$$

$$\leq \underbrace{W_\theta(\boldsymbol{u}(P^*)) - W_\theta(\hat{\boldsymbol{u}}(P^*))}_{=C_1} + \underbrace{W_\theta(\hat{\boldsymbol{u}}(\hat{P})) - W_\theta(\boldsymbol{u}(\hat{P}))}_{=C_2}$$

Let $B_1(\hat{\mu}) = \max_{i \in [\![n]\!]} \psi'(\hat{\boldsymbol{u}}_i(P^*), \alpha)$.

We first prove:

$$C_1 \leq 2B_1(\hat{\mu}) \sqrt{n \|v\|_2^2} \sqrt{\sum_{(i,j) \in [\![n]\!]^2} (\hat{\mu}_{ij} - \mu_{ij})^2}. \tag{2}$$

To prove (2), we start by using the concavity of $\psi(., \alpha)$ for $\alpha \leq 1$. Let $\Phi(.) = \frac{1}{2}\psi(., \alpha)$. We have:

$$C_1 = \sum_{i=1}^n \big( \Phi(\boldsymbol{u}(P^*)) - \Phi(\hat{\boldsymbol{u}}(P^*)) \big) \leq \sum_{i=1}^n \Phi'(\hat{\boldsymbol{u}}_i(P^*)) \big( \boldsymbol{u}(P^*)) - \hat{\boldsymbol{u}}(P^*) \big)$$

$$\text{thus } C_1 \leq \sum_{i=1}^n \sum_{j=1}^n \Phi'(\hat{\boldsymbol{u}}_i(P^*))(\mu_{ij} - \hat{\mu}_{ij})(P_{ij}^* + P_{ji}^*)v$$

$$= \sum_{i=1}^n \sum_{j=1}^n \big( \underbrace{\Phi'(\hat{\boldsymbol{u}}_i(P^*))(\mu_{ij} - \hat{\mu}_{ij}) + \Phi'(\hat{\boldsymbol{u}}_j(P^*))(\mu_{ji} - \hat{\mu}_{ji})}_{=A_{ij}} \big) P_{ij}^* v$$

where, similarly to the proof of Lemma 1, we swapped the indexed $(i, j)$ in the $\Phi'(\hat{\boldsymbol{u}}_i(P^*))\mu_{ij}P_{ji}^* v$, which is possible because $i$ and $j$ span the same range in the sum.

Notice that the terms $A_{ij}P_{ij}^* v$ are all zero except if $i \in \mathcal{N}$ and $j \in \mathcal{I}$ (because $P_{ijk}^* = 0$ otherwise). For $i \in \mathcal{N}$, let $\sigma_i$ be a ranking which ranks $(A_{ij})_{j \in \mathcal{I}}$ in decreasing order, i.e., $\sigma_i(j) < \sigma_i(j') \implies A_{ij} \geq A_{ij'}$. Using Lemma 3 in Appendix E, we have:

$$C_1 \leq \max_{P \in \mathcal{P}} \sum_{i \in \mathcal{N}}^n \sum_{j \in \mathcal{I}} A_{ij}P_{ij}v = \sum_{i \in \mathcal{N}}^n \sum_{j \in \mathcal{I}} A_{ij}v_{\sigma_i(j)}$$

Now let $V = [v_{\sigma_i(j)}]_{\substack{i \in \mathcal{N} \\ j \in \mathcal{I}}}$. By Cauchy-Shwarz inequality and denoting $\|X\|_F = \sqrt{\sum_{ij} X_{ij}^2}$ the Frobenius norm of matrix $X$, we have $\|V\|_F = \sqrt{n \|v\|_2^2}$ and $\|A\|_F \leq B_1(\hat{\mu})(\|\mu - \hat{\mu}\|_F +$

---

[12]The dependency on $\hat{\mu}$ in $B(\hat{\mu})$ is because $\psi'(., \alpha)$ is not bounded in general. In practice, we use $\psi(x + \eta, \alpha)$ for a small $\eta > 0$ to avoid the singular point at 0, in which case $B < \psi'(\eta, \alpha)$.

$\|\mu^\top - \hat{\mu}^\top\|_F$), leading to:

$$C_1 \leq \sqrt{n \|v\|_2^2} \|A\|_F \leq 2B_1(\hat{\mu})\sqrt{n \|v\|_2^2} \|\mu - \hat{\mu}\|_F$$

which proves (2).

Similarly, using $B_2(\hat{\mu}) = \max_{i \in [\![n]\!]} \psi'(u_i(\hat{P}), \alpha)$ and the same arguments as above, we obtain:

$$C_2 \leq \sqrt{n \|v\|_2^2} \|A\|_F \leq 2B_2(\hat{\mu})\sqrt{n \|v\|_2^2} \|\mu - \hat{\mu}\|_F$$

which yields the desired result. □

## D  Comparison to utility/inequality trade-offs

In this appendix, we provide the proofs of Section 3, and describe more precisely how we applied quality-weighted exposure and equality of exposure in reciprocal recommendation.

### D.1  One-sided recommendation: quality-weighted exposure

We prove here Proposition 2 of Section 3. The result shows that in some cases, compared to any choice of the parameter $\theta \in \Theta$ of the welfare approach, quality-weighted exposure leads to the undesirable behavior of *decreasing user utility* while *increasing inequalities of exposure* between items. Figure 5 gives an example.

**Proposition 2.** *The following claims hold irrespective of the choice of $\boldsymbol{u}^{\mathrm{qua},\beta} \in \mathcal{U}_\beta^{\mathrm{qua}}$.*

*For every $d \in \mathbb{N}_*$ and every $N \in \mathbb{N}_*$, there is a one-sided recommendation problem, with $d+1$ items and $N(d+1)$ users, such that $\forall \theta \in \Theta$, we have:*

$$\left(\exists \beta > 0, \boldsymbol{u}_\mathcal{N}^\theta \succ_\mathrm{L} \boldsymbol{u}_\mathcal{N}^{\mathrm{qua},\beta} \text{ and } \boldsymbol{u}_\mathcal{I}^\theta \succ_\mathrm{L} \boldsymbol{u}_\mathcal{I}^{\mathrm{qua},\beta}\right) \qquad and \qquad \lim_{\beta \to \infty} \frac{\sum_{i \in \mathcal{N}} u_i^{\mathrm{qua},\beta}}{\sum_{i \in \mathcal{N}} u_i^\theta} \xrightarrow{d \to \infty} \frac{5}{6}.$$

*Proof.* We prove it for $N = 1$, the more general case is just obtained by repeating the pattern with $d+1$ items and $d+1$ users.

Let $i_1, ..., i_{d+1}$ be the indexes of the users and $j_1, ..., j_{d+1}$ the indexes of the items. The preferences have the following pattern:

$$\forall k \in [\![d+1]\!], \mu_{i_k j_k} = 1 \qquad\qquad \forall k \in [\![d]\!], \mu_{i_k j_{d+1}} = \frac{1}{2}$$

all other $\mu_{ij}$ (for user $i$ and item $j$) are set to $0$ (note that we are in a problem with one-sided preferences, which means $\mu_{ji} = 1$ for every item $j$ and user $i$.

We consider a task with a single recommendation slot ($v_1 = 1, v_2 = ... = v_{|\mathcal{I}|} = 0$). On that problem, the optimal ranking for every $\theta \in \Theta$ is to show item $j_k$ to user $i_k$, which leads to perfect equality in terms of item exposure, and maximizes every user utility. It is thus leximin optimal for both users and items for every $\theta \in \Theta$.

Then, the qualities are equal to:

$$\forall k \in [\![d]\!], q_{j_k} = 1 \qquad\qquad q_{j_{d+1}} = \frac{1}{2}d + 1$$

the target exposure is thus $t_{j_k} = \frac{d+1}{\frac{3}{2}d+1}$ for $k \in [\![d]\!]$ and $t_{j_{d+1}} = (d+1)\frac{\frac{1}{2}d+1}{\frac{3}{2}d+1}$.

Since the problem is symmetric in the users $i_1, ..., i_d$, by the concavity of $F_\beta^{\mathrm{qua}}(\boldsymbol{u}(P))$ with respect to $P$, there is an optimal ranking described by a single probability $p$ as:

$$\forall k \in [\![d]\!], P_{i_k j_k} = 1 - p \qquad P_{i_k j_{d+1}} = p \qquad P_{i_{d+1} j_{d+1}} = 1$$

Note that for such a $P$, $\forall k \in [\![d]\!], u_{i_k}^{\mathrm{qua},\beta}(P) = 1 - \frac{1}{2}p$, and it is clearr that there is $\beta > 0$ such that $p > 0$, which then implies $\boldsymbol{u}^\theta \succ_\mathrm{L} \boldsymbol{u}_\mathcal{N}^{\mathrm{qua},\beta}$ and $\boldsymbol{u}^\theta \succ_\mathrm{L} \boldsymbol{u}_\mathcal{I}^{\mathrm{qua},\beta}$.

Now, as $\beta \to \infty$, $p$ is such that exposure equals its target, which leads to the following equation:

$$dp + 1 = (d+1)\frac{\frac{1}{2}d + 1}{\frac{3}{2}d + 1}.$$

We thus get $p = \frac{d+1}{d}\frac{d+2}{3d+2} - \frac{1}{d} \xrightarrow[d\to\infty]{} \frac{1}{3}$, which gives the result $u_{i_k}^{\text{qua},\beta}(P) = 1 - \frac{1}{2}p \xrightarrow[p\to\frac{1}{3}]{} \frac{5}{6}$.

Notice that similarly to Proposition 3, the result does not depend on the choice of $\boldsymbol{u}^{\text{qua},\beta}$ because the sum of user utilities converges. $\qquad\square$

## D.2 Reciprocal recommendation: equality of exposure

We now prove Proposition 3.

**Proposition 3.** *For $\beta > 0$, let $\mathcal{U}_\beta^{\text{eq}} = \arg\max_{\boldsymbol{u}\in\mathcal{U}} F_\beta(\boldsymbol{u})$. The claim below holds irrespective of the choice of $\boldsymbol{u}^{\text{eq},\beta} \in \mathcal{U}_\beta^{\text{eq}}$. Let $n \geq 5$. There is a reciprocal recommendation task with $n$ users such that:*

$$\forall \theta \in \Theta, \boldsymbol{u}^\theta, \exists \beta > 0: \quad \forall i \in [\![n]\!], u_i^\theta > u_i^{\text{eq},\beta} \qquad \text{and} \qquad \lim_{\beta\to\infty}\sum_{i\in\mathcal{N}}u_i^{\text{eq},\beta} = 0.$$

*Proof.* The example is given in Figure 5. We still consider a recommendation task with a single recommendation slot.

Let us rename the users by $i_1, i_2, ..., i_5$. The preference patterns are $\mu_{i_1 i_2} = \mu_{i_1 i_3} = 1$ and $\mu_{i_4 i_5} = 1$. Apart from $\mu_{ij} = \mu_{ji}$, other $\mu_{ij}$s are 0. In this proof, we show that $u_{i_1}^{\text{eq},\beta} = 2u_{i_2}^{\text{eq},\beta}$ for every $\beta$, which implies that $u_{i_1}^{\text{eq},\beta} \xrightarrow[\beta\to\infty]{} 0$ because 0 utility for every user is feasible. On this task, the leximin ranking also maximizes the sum of users utilities (as shown in Figure 5), so the optimal ranking is the same for every $\theta \in \Theta$, and every user has a two-sided utility of at least 1.5.

Since $F_\beta(\boldsymbol{u})$ is strictly Schur-concave for $\beta > 0$, $i_2$ and $i_3$ always have the same utility in an optimal utility profile (because they play a symmetric role). $i_4$ and $i_5$ also have the same utility. Note that the interest of $i_4$ and $i_5$ in that problem is to make it possible to recommend them to $i_1$, which has 0 value.

Similarly to the problem in one-sided recommendation, the only way to decrease the penalty is to reduce the utility of $i_1, i_4, i_5$. However, reducing the utility of $i_1$ can only be done by either recommending $i_4$ or $i_5$ to $i_1$, or recommending $i_4/i_5$ to $i_2/i_3$. In all cases, decreasing $i_1$'s utility decreases $i_2/i_3$'s utilities.

More precisely, because of the symmetries and the concavity of $F_\beta(\boldsymbol{u}(P))$ with respect to $P$, for every $\beta > 0$, there is an optimal ranking tensor described by three probabilities $p, q, q'$ such that:[13]

$$P_{i_1 i_2} = P_{i_1 i_3} = \frac{1}{2}p \qquad P_{i_2 i_1} = P_{i_3 i_1} = q \qquad P_{i_4 i_5} = P_{i_5 i_4} = q'$$

$$P_{i_1 i_4} = P_{i_1 i_5} = \frac{1}{2}(1-p) \quad P_{i_2 i_3} = P_{i_2 i_4} = P_{i_2 i_5} = \frac{1}{3}(1-q) \quad P_{i_4 i_1} = P_{i_4 i_2} = P_{i_4 i_3} = \frac{1}{3}(1-q')$$

$$P_{i_3 i_2} = P_{i_3 i_4} = P_{i_3 i_5} = \frac{1}{3}(1-q) \quad P_{i_5 i_1} = P_{i_5 i_2} = P_{i_5 i_3} = \frac{1}{3}(1-q')$$

In all cases, the two-sided utility are

$$u_{i_1}(P) = \underbrace{p}_{\substack{P_{i_1 i_2}\mu_{i_1 i_2}+P_{i_1 i_3}\mu_{i_1 i_3} \\ \text{user-side utility}}} + \underbrace{2q}_{\substack{P_{i_2 i_1}\mu_{i_2 i_1}+P_{i_3 i_1}\mu_{i_3 i_1} \\ \text{item-side utility}}} \qquad \text{and} \qquad u_{i_2}(P) = q + \frac{1}{2}p$$

Thus, in an optimal ranking for $F_\beta(\boldsymbol{u})$, we must have $u_{i_1}(P) = 2u_{i_2}(P)$. Equality, which is achieved at $\beta \to \infty$ can then only be at 0 utility for every user (since 0 is feasible).

The task used in the proof contains only 5 users. Any number of users can be added to the group $\{i_4, i_5\}$, with a "complete" preference profile ($\mu_{ij} = 1$ for all pair $i, j$ in that group). $\qquad\square$

---

[13]Since there is a single recommendation slot, we identify $P_{ij1}$ with $P_{ij}$

The Lorenz efficiency of our welfare approach guarantees that it cannot exhibit the undesirable behaviors of equality or quality-weighted exposure penalties described in Propositions 2 and 8.

### D.3 Equality of exposure and quality-weighted exposure in reciprocal recommendation

In one-sided recommendation with one-sided preferences, equality of exposure is the same as equality of utility. More generally, let $e_j(P) = \sum_{i \in \mathcal{N}} P_{ij} v$ the total exposure of item $j$. Equality of exposure is defined by:

$$F_\beta^{\mathrm{expo}}(P) = \sum_{i \in \mathcal{N}} \bar{u}_i(P) - \beta \sqrt{\sum_{j \in \mathcal{I}} \left( e_j(P) - \frac{|\mathcal{N}|}{|\mathcal{I}|} \|v\|_1 \right)^2}$$

In one-sided recommendation, parity of exposure is relatively well behaved because the exposure target $\frac{|\mathcal{N}|}{|\mathcal{I}|} \|v\|_1$ is constant. Driving towards equality can thus not lead to a decrease of the total exposure budget, which was the problem with equality of utility in settings with two-sided preferences (driving towards equality of utility leads to a decrease of total utility), as we described in Section 3.

The formula allows us to extend parity of exposure in the next section and in our experiments, since it is also valid in reciprocal recommendation. Likewise, the formula of quality-weighted exposure that is also valid in reciprocal recommendation is given by:

$$F_\beta^{\mathrm{qua}}(P) = \sum_{i \in \mathcal{N}} \bar{u}_i(P) - \beta \sqrt{\sum_{j \in \mathcal{I}} \left( e_j(P) - \frac{q_j E}{Q} \right)^2}.$$

The result below shows that equality of exposure and quality-weighted exposure lead to inefficiencies in reciprocal recommendation settings:

**Proposition 8.** *For every $n \in \mathbb{N}_*$, there is a reciprocal recommendation task with $n$ users such that:*

$$\forall \theta \in \Theta, \exists \beta > 0: \qquad \boldsymbol{u}^\theta \succ_{\mathrm{L}} \boldsymbol{u}^{\mathrm{expo},\beta} \qquad and \qquad \boldsymbol{u}^\theta \succ_{\mathrm{L}} \boldsymbol{u}^{\mathrm{qua},\beta}.$$

*Moreover,* $\displaystyle \lim_{\beta \to \infty} \sum_{i \in \mathcal{N}} u_i^{\mathrm{expo},\beta} = \frac{2}{n} \sum_{i \in \mathcal{N}} u_i^\theta$ *and* $\displaystyle \lim_{\beta \to \infty} \sum_{i \in \mathcal{N}} u_i^{\mathrm{qua},\beta} = \frac{2+n}{2n} \sum_{i \in \mathcal{N}} u_i^{\mathrm{sum}}.$

*Proof.* An example of extreme case is with $n$ users when there is a "leader" who is the only possible match with other users. We consider a single recommendation slot. The preferences are:

$$\forall j \in \{2, \ldots, n\}, \mu_{1j} = \mu_{j1} = 1 \qquad \forall (i,j) \in \{2, \ldots, n\}^2, \mu_{ij} = 0.$$

On this task, the for every $\theta \in \Theta$, the optimal ranking is given by:

$$\forall j \in \{2, \ldots, n\}, P_{1j} = \frac{1}{n-1} \qquad \forall i \in \{2, \ldots, n\} P_{i1} = 1.$$

The reason it is the only possible optimal ranking is because it is leximin optimal and has the maximum achievable sum of utilities. The utilities are then $u_1(P) = n$ and $u_i(P) = 1 + \frac{1}{n-1}$, which leads to $\sum_{i=1}^n u_i = 2n$.

**Equality of exposure** Driving towards equality of exposure requires to reduce the exposure of user 1, which in turn reduces the utility of user 1 and the utilities of those who user 1 is less exposed to. Thus, there is $\beta > 0$ such that $\boldsymbol{u}^\theta \succ_{\mathrm{L}} \boldsymbol{u}^{\mathrm{expo},\beta}$ because of the loss of efficiency. Finally, by the concavity of the objective with respect to $P$, and by the symmetry of the problem with respect to $i_2, \ldots, i_n$, we can conclude that an optimal way to achieve perfect equality of exposure is to recommend, to every user $i$, every user $j \neq i$ with probability $\frac{1}{n-1}$. The utility is then $u_1(P) = 1 + (n-1)\frac{1}{n-1}$ and $u_i(P) = \frac{2}{n-1}$ for $i \geq 2$, leading to $\sum_{i=1}^n u_i = 4$, which gives the result.

**Quality-weighted exposure** On the same example, the qualities are $q_1 = n - 1$ and $q_i = 1$ for $1 \geq 2$. The total exposure targets are then $t_1 = \frac{1}{2}n$ and $t_i = \frac{n}{2(n-1)}$. These exposure targets mean less exposure for 1 than in the leximin ranking. Thus $\beta$ sufficiently large has the effect of reducing 1's exposure[14], which reduces the utility of 1 and the users to whom 1 is less recommended. Thus

---

[14]Direct calculations of the derivatives show that when $\beta > 0$ is too small the penalty has no effect.

Figure 5: **Left:** Example of a reciprocal recommendation task where equality of utility leads to 0 utility (see the proof of Prop. 3 in App. D). There is one recommendation slot per user. We give the recommendation probabilities and utilities for the utilitarian ranking and three users, the other ones are obtained by the symmetry of the problem. The utilitarian ranking is also leximin optimal, so our approach yields the same recommendations for all $\theta$. **Right:** Example where quality-weighted exposure reduces user utility while increasing inequalities between items.

$\boldsymbol{u}^{\theta} \succ_{\mathrm{L}} \boldsymbol{u}^{\mathrm{qua},\beta}$. By the symmetry of the problem, as $\beta \to \infty$, quality weighted exposure is achieved by setting:

$$\forall j \in \{2,\ldots,n\} : P_{1j} = \frac{1}{n-1} \qquad\qquad P_{j1} = \frac{n}{2(n-1)}$$

$$\forall j' \in \{2,\ldots,n\}, j' \neq j, P_{jj'} = \frac{1 - \frac{n}{2(n-1)}}{n-2} = \frac{1}{2(n-1)}$$

The utilities are then $u_1(P) = 1 + (n-1)\frac{n}{2(n-1)} = 1 + \frac{n}{2}$ and $u_i(P) = \frac{n}{2(n-1)} + \frac{1}{n-1} = \frac{n+2}{2(n-1)}$. The total utility is thus $2 + n$, which gives the result. $\qquad\square$

The Lorenz efficiency of our welfare approach guarantees that it cannot exhibit the undesirable behaviors of parity or quality-weighted exposure penalties described in Propositions 2 and 8.

# E  A generic Frank-Wolfe algorithm for ranking

In this section, we present a general form of our algorithm presented in Section 4, as well as the proofs of the claims.

Let $F : \mathbb{R}^n \to \mathbb{R}$, concave, and we want to find

$$P^* \in \operatorname*{argmax}_{P \in \mathcal{P}} F(\boldsymbol{u}(P)). \tag{3}$$

Let $\langle X \mid Y \rangle = \sum_{ijk} X_{ijk} Y_{ijk}$ be the dot product between three-way tensors, and let $\nabla(F \circ \boldsymbol{u})(P)$ be the gradient of $P \mapsto F(\boldsymbol{u}(P))$ taken at $P$, i.e., $(\nabla(F \circ \boldsymbol{u}))_{ijk} = \frac{\partial F \circ \boldsymbol{u}}{\partial P_{ijk}}$

Starting from $P^{(0)} \in \mathcal{P}$ (in our experiments we always use a utilitarian ranking $P^{(0)} \in \operatorname{argmax}_{P \in \mathcal{P}} \sum_{i=1}^{n} u_i(P)$), the Frank-Wolfe algorithm alternates two steps for $t \geq 1$:

1. let $\tilde{P} \in \operatorname{argmax}_{P \in \mathcal{P}} \langle P \mid \nabla(F \circ \boldsymbol{u})(P^{t-1}) \rangle$
2. $P^{(t)} = (1 - \gamma^{(t)})P^{(t-1)} + \gamma^{(t)}\tilde{P}$ with $\gamma^{(t)} = \frac{2}{t+2}$

The stepsize $\frac{2}{t+2}$ is from Clarkson [13, Section 3], which avoids a line search and in our experiments seemed to yield acceptable results. Irrespective of the step size, the fundamental results which allows to use Frank-Wolfe in the setting of (3) are the two following lemmas:

**Lemma 2.** *Recall that $u_i(P) = \sum_{i=1}^{n} \mu_{ij}(P_{ij} + P_{ji})v$. Let $\frac{\partial F}{\partial u_i}$ denote the derivative of $F$ with respect to its $i$-th argument and $\frac{\partial F}{\partial u_i}(\boldsymbol{u}(P))$ the value of this derivative at $\boldsymbol{u}(P)$.*

*Then, $\forall i \in \mathcal{N}, \forall j \in \mathcal{I}, \forall k \in [\![|\mathcal{I}|]\!]$, we have:*

$$\frac{\partial F \circ \boldsymbol{u}}{\partial P_{ijk}}(P) = \left(\mu_{ij}\frac{\partial F}{\partial u_i}(\boldsymbol{u}(P)) + \mu_{ji}\frac{\partial F}{\partial u_j}(\boldsymbol{u}(P))\right)v_k.$$

*Proof.* The result is a consequence of the chain rule:

$$\frac{\partial F \circ \boldsymbol{u}}{\partial P_{ijk}}(P) = \sum_{p=1}^{n} \frac{\partial F}{\partial u_p}(\boldsymbol{u}(P))\frac{\partial u_p(P)}{\partial P_{ijk}}$$

With

$$u_p(P) = \sum_{q=1}^{n} \mu_{pq} \sum_{r=1}^{|\mathcal{I}|} (P_{pqr} + P_{qpr})v_k.$$

Thus $\frac{\partial u_p(P)}{\partial P_{ijk}} = (\mu_{ij}\mathbb{1}_{\{p=i\}} + \mu_{ji}\mathbb{1}_{\{p=j\}})v_k$, which gives the desired result. $\qquad\square$

**Lemma 3.** *Let $A$ be an $n \times n$ matrix with $A_{ij} \in \mathbb{R}$ (not necessarily non-negative). Let $v \in \mathbb{R}^{|\mathcal{I}|}$ with non-negative and non-increasing entries, i.e., $\forall k \in [\![|\mathcal{I}| - 1]\!]$, $v_k \geq v_{k-1} \geq 0$. Let $K$ be the last index such that $v_K > 0$ (or $K = |\mathcal{I}|$ if there is no such index).*

*Let $P \in \mathcal{P}$ such that:*

$$\forall i, \forall \sigma_i \in \mathfrak{S}(P_i), \forall (j, j') \in \mathcal{I}^2 : \ \Big(\sigma_i(j) \leq K \text{ and } \sigma_i(j) < \sigma_i(j') \implies A_{ij} \geq A_{ij'}\Big).$$

*And let $X$ be the $n \times n \times |\mathcal{I}|$ tensor defined as $X_{ijk} = A_{ij}v_k$.*

*Then $P \in \mathrm{argmax}_{P \in \mathcal{P}}\langle P \,|\, X \rangle$.*

*Moreover, if $\forall k \in [\![|\mathcal{I}| - 1]\!]$, $v_k > v_{k-1} \geq 0$, then for every $P \in \mathrm{argmax}_{P \in \mathcal{P}}\langle P \,|\, X \rangle$, we have:*

$$\forall i, \forall \sigma_i \in \mathfrak{S}(P_i), \forall (j, j') \in \mathcal{I}^2 : \ \Big(\sigma_i(j) < \sigma_i(j') \implies A_{ij} \geq A_{ij'}\Big).$$

*Proof.* The result stems from the rearrangement inequality (also known as the Hardy-Littlewood inequality [23]), which states that for two vectors $a \in \mathbb{R}_+^n$, and $b \in \mathbb{R}^n$, $\mathrm{argmax}_\nu \sum_{j=1}^{n} a_{\nu(j)}b_j$, where $\nu$ spans the permutations of $[\![n]\!]$, is the set of permutations such that $b$ is ordered similarly to $(a_{\nu(i)})_{i=1}^{n}$. If the $a_k$s are non-increasing, then every permutation that sorts $b$ in decreasing order is in the argmax. We need the reciprocal statement for the second part of our Lemma: if the $a_i$s are strictly decreasing, then only the permutations that sort $b$ in decreasing order are in $\mathrm{argmax}_\nu \sum_{j=1}^{n} a_{\nu(j)}b_j$. Note that these arguments are well-known in learning to rank [see, e.g., 14].

In our case, notice that

$$\langle P \,|\, X \rangle = \sum_{i \in \mathcal{N}} \Big( \sum_{j \in \mathcal{I}} A_{ij}P_{ijk}v_k \Big)$$

The maximization over $P$ can then be performed over each user $i$ (and thus each bistochastic matrix $P_i$ separately). Now, if $P_i$ is such that every $\sigma_i \in \mathfrak{S}(P_i)$ orders $A_{ij}$ in decreasing order, then by the rearrangement inequality $\sigma_i \in \mathrm{argmax}_\nu \sum_{j \in \mathcal{I}} A_{ij}v_{\nu(j)}$. Notice that if only the $K$ first elements of $v$ are non-zero, we only need a top-$K$ ranking. This gives us the first part of the thoerem.

The second part of the theorem follows from the reciprocal of the rearrangement inequality, since for $P_i$ to be an optimal stochastic ranking for $\sum_{j \in \mathcal{I}} A_{ij}P_{ijk}v_k$, every permutation $\sigma_i$ in its support must be in $\mathrm{argmax}_\nu \sum_{j \in \mathcal{I}} A_{ij}v_{\nu(j)}$. $\qquad\square$

## E.1 Proof of Theorem 1

Lemma 2 and 3 together are sufficient to give algorithms for the inference of stochastic rankings using our welfare function (1) and using the penalties of Section 3, by computing the partial derivatives $\frac{\partial F}{\partial u_i}$. The main result of Section 4, which we prove now, instantiates this principle for the welfare function approach:

**Theorem 1.** *Let $\tilde{\mu}_{ij} = \Phi_i'\big(u_i(P^{(t)})\big)\mu_{ij} + \Phi_j'\big(u_j(P^{(t)})\big)\mu_{ji}$. Let $\tilde{P}$ such that:*

$$\forall i \in \mathcal{N}, \forall \tilde{\sigma}_i \in \mathfrak{S}(\tilde{P}_i): \ \ \tilde{\sigma}_i(j) < \tilde{\sigma}_i(j') \implies \tilde{\mu}_{ij} \geq \tilde{\mu}_{ij'}. \textit{ Then } \tilde{P} \in \mathop{\mathrm{argmax}}_{P \in \mathcal{P}}\langle P \,|\, \nabla W(P^{(t)})\rangle.$$

*Proof.* Notice that with $W(P) = F(\boldsymbol{u}(P)) = \sum_{i=1}^{n} \Phi_i(u_i(P))$, then $\frac{\partial F}{\partial u_i}(\boldsymbol{u}(P)) = \Phi_i'(u_i(P))$. By Lemma 2, we have that $\langle P \,|\, \nabla F(P^{(t)})\rangle$ is of the form $\langle P \,|\, X \rangle$ with $X_{ijk} = A_{ij}v_k$ with $A_{ij} = \tilde{\mu}_{ij}$, so the result is implied by Lemma 3. $\qquad\square$

## E.2  Proof of Proposition 4

**Proposition 4.** *Let* $B = \max\limits_{i \in [\![n]\!]} \|\Phi_i''\|_\infty$ *and* $U = \max\limits_{\boldsymbol{u} \in \mathcal{U}} \|\boldsymbol{u}\|_2^2$. *Let* $K$ *be the maximum index of a nonzero value in* $v$ *(or* $|\mathcal{I}|$*). Then* $\forall t \geq 1, W(P^{(t)}) \geq \max\limits_{P \in \mathcal{P}} W(P) - O(\frac{BU}{t})$. *Moreover, for each user, an iteration costs* $O(|\mathcal{I}| \ln K)$ *operations and requires* $O(K)$ *additional bytes of storage.*

*Proof.* Note that $\mathcal{P}$ is a simplex over ranking tensors containing one deterministic ranking for each user. Using [13, Section 3], the Frank-Wolfe algorithm with our step-size converges in $O\big(\frac{C_W}{t}\big)$, where, using [13, Equation 11] and denoting by $\nabla^2 W$ the Hessian of $W$, we have

$$C_W \leq \sup_{\substack{\boldsymbol{u}, \boldsymbol{u'} \in \mathcal{U} \\ \tilde{\boldsymbol{u}} \in \mathcal{U}}} -\frac{1}{2}(\boldsymbol{u} - \boldsymbol{u'})^\top \nabla^2 W(\tilde{\boldsymbol{u}})(\boldsymbol{u} - \boldsymbol{u'}) \leq \frac{B}{2} \sup_{\boldsymbol{u}, \boldsymbol{u'} \in \mathcal{U}} \|\boldsymbol{u} - \boldsymbol{u'}\|_2^2 \leq 2BU.$$

where we used $\|\boldsymbol{u} - \boldsymbol{u'}\|_2^2 \leq 2\|\boldsymbol{u}\|_2^2 + 2\|\boldsymbol{u'}\|_2^2$.

For the computation cost, we use Lemma 3, which is more precise than Theorem 1, to see that finding the argmax only requires a top-$K$ ranking. While technically any $P \in \mathcal{P}$ should contain a whole bistochastic matrix, it is not necessary to store a completion of the top-$K$ rankings because they have no impact on the utility. As such, storing each $\tilde{P}$ only costs $O(K)$ bytes per user, which contain the indices of the top-K items in the ranking found by Theorem 2.

Computing the two-sided utilities costs $O(|\mathcal{N}||\mathcal{I}|)$, and thus $O(|\mathcal{I}|)$ per user. Moreover, computing the top-K ranking costs $O(|\mathcal{I}| \ln K)$ in the worst case, with a streaming method that maintains a min-heap of the top-K elements seen so far, and finish with sorting the top-$K$ elements. $\qquad \square$

Notice that for faster average performance, the top-K sort can be performed using a fast selection algorithm (such as quickselect), to obtain the top-$K$ elements with $O(|\mathcal{I}|)$ expected time complexity, and then sorting, yielding $O(|\mathcal{I}| + K \ln K)$ expected time complexity per user at each iteration.

# F   Additional experimental results

Our experiments are fully implemented in Python 3.9 using PyTorch[15]. We provide the code as supplementary material. We compare our welfare maximization approach with the fairness penalties presented in Section 3.

We also compare ourselves to the algorithm FairRec from Patro et al. [47] (referred to as *Patro et al.* in the figures and description), who consider envy-freeness as user-side fairness criterion, and max-min share of exposure as item-side fairness criterion. Envy-freeness states that every user should prefer their recommendation list to that of any other user. The max-min exposure criterion on the item side means that each user should receive an exposure of at least $\beta \frac{E}{|\mathcal{I}|}$, where $\beta$ is a parameter allowing to control how much exposure is guaranteed to items. We vary this parameter in our experiments to show the trade-offs achieved by *Patro et al.*. Since *Patro et al.* does not produce rankings, we took the recommendation list with the given order as a ranked list.

## F.1   One-sided recommendation: Lastfm-2k dataset

We describe in this section the details of the experiments presented in Section 5.1. We use a dataset from the online music service Last.fm[16]. In the main paper, we presented results on **Lastfm-2k** from Cantador et al. [9] which contains real play counts of $2k$ users for $19k$ artists, and was used by Patro et al. [47] who also study two-sided fairness in recommendation. We filter the top $2,500$ items most listened to. Following [32], we pre-process the raw counts with $\log$-transformation. We split the dataset into train/validation/test sets, each including $70\%/10\%/20\%$ of the user-item play counts. We create three different splits using three random seeds. One-sided preferences are estimated using the standard matrix factorization algorithm[17] of Hu et al. [29] trained on the train set, with hyperparameters selected on the validation set by grid search. The number of latent factors is chosen in $[16, 32, 64, 128]$, the regularization in $[0.1, 1., 10., 20., 50.]$, and the confidence weighting parameter in $[0.1, 1., 10., 100.]$. The estimated preferences we use are the positive part of the resulting estimates.

Rankings are inferred from these estimated preferences. The exposure weights we use in the computation of utilities are the standard weights of the *discounted cumulative gain* (DCG) (also used in e.g., [54, 7, 42]): $\forall k \in [\![|\mathcal{I}|]\!], v_k = \frac{1}{\log_2(1+k)}$. For each ranking approach, the Frank-Wolfe algorithm is run with $5000$ iterations to make sure we are close to convergence, and the number of recommendation slots is set to $40$.

We evaluate rankings on estimated preferences, considered as ground truth, following many works on fair recommendation [54, 47, 60, 63]. This is because the goal is to evaluate the fairness of ranking algorithms themselves, rather than biases in preference estimates. All results are averaged over three random seeds. To obtain various trade-offs, for *welf* we vary $\lambda$ in $[0.001, 0.01, 0.05, 0.075, 0.1, 0.125, 0.15, 0.2, 0.3, 0.325, 0.35]$ and $[0.4, 0.45, 0.5, 0.55, 0.6, 0.65, 0.7, 0.75, 0.8, 0.9, 0.95, 0.99, 0.999]$. For *Patro et al.* we vary $\beta$ in $[0.01, 0.05, 0.1, 0.15, 0.2, 0.25, 0.3, 0.35, 0.4]$ and $[0.45, 0.5, 0.55, 0.6, 0.65, 0.7, 0.75, 0.8, 0.85, 0.9, 0.95, 1]$, and for other methods we vary $\beta$ in $[0.001, 0.005, 0.01, 0.015, 0.0175, 0.02, 0.025, 0.03, 0.035, 0.04, 0.045, 0.05, 0.055, 0.06]$ and $[0.065, 0.07, 0.075, 0.08, 0.085, 0.09, 0.095, 0.1, 0.105, 0.11, 0.2, 0.5, 1, 2, 5, 10, 20, 30, 40, 50, 70, 100]$.

**Item-side fairness**   Figure 6 presents the various trade-offs achieved by each method in one-sided recommendation, as discussed in Section 5.1. We observe that only *qua.-weighted* is unable to reach equal exposure because of its quality-weighted exposure target: perfectly equal exposure is only permitted when all items have the same quality.

**Two-sided fairness**   Figure 7 shows the effect of varying $\alpha_1$ and $\lambda$ on user fairness as in Figure 3 of the main paper, but with results repeated over three random seeds. We observe the same trade-offs and conclude again that *welf* is better than *Patro et al.* and *eq. exposure*, in terms of its impact on worse-off users.

---

[15]http://pytorch.org
[16]https://www.last.fm/
[17]Using the Python library Implicit: https://github.com/benfred/implicit (MIT License).

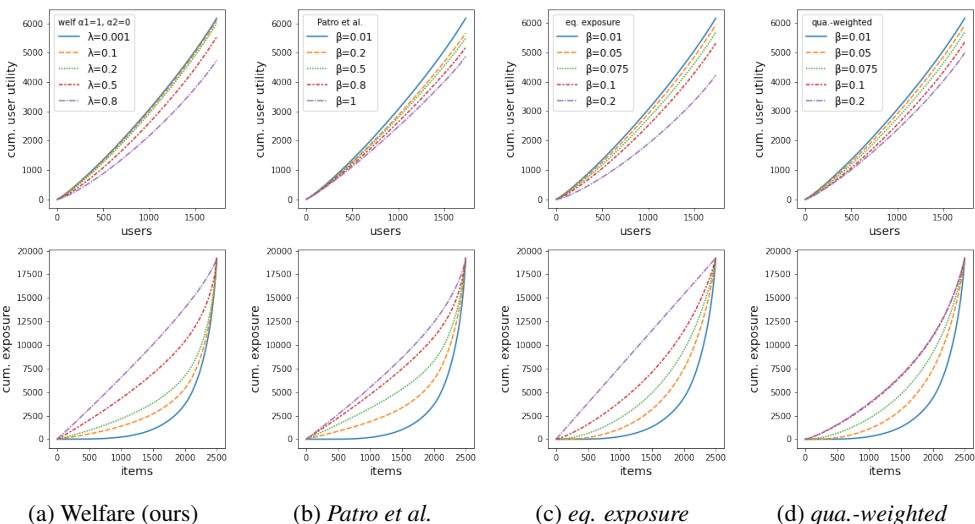

(a) Welfare (ours)   (b) *Patro et al.*   (c) *eq. exposure*   (d) *qua.-weighted*

Figure 6: representative trade-offs achieved by the various compared methods on Lastfm-2k. The trade-offs achieved by the different methods look alike, except that *qua.-weighted* does not aim at reaching equality of exposure for exteme values of $\beta$. See Section 5.1 for the discussions on the differences between the trade-offs achieved by the different approaches.

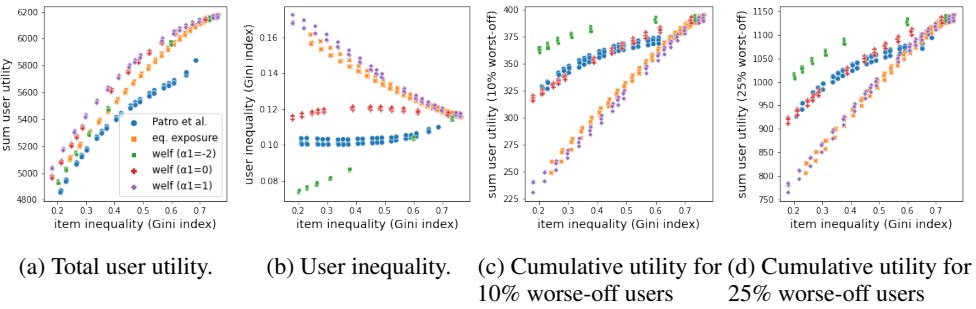

(a) Total user utility.   (b) User inequality.   (c) Cumulative utility for 10% worse-off users   (d) Cumulative utility for 25% worse-off users

Figure 7: Focus on user fairness on Lastfm-2k: effect of varying $\alpha_1$ (user-side curvature of the welfare function) keeping $\alpha_2 = 0$. The figure shows all the results obtained with a repetition of three seeds. Overlapping points correspond to the same model parameter across different seeds. We can see that the variance is negligible compared to the observed differences.

**The importance of considering the whole Lorenz curve** In Fig. 8 we show the results of the same models as before, but changing the way we measure the item inequality: using the standard deviation of exposure rather than the Gini index. Now, *eq. exposure* dominates the total utility/item inequality plot, since the plot corresponds exactly to the objective function of the algorithm. Comparing *eq. exposure* with *welf* $\alpha_1 = 1$, we now see that the trade-offs are different, with *eq. exposure* performing better on the worse-off users. Comparing *welf* $\alpha_1 = 0$ and *Patro et al.*, we see that they still exhibit similar behaviors, with *welf* $\alpha_1 = 0$ being better for better off users. Finally, *welf* $\alpha_1 = -2$ still dominates the othe methods in terms of performance on the worse-off users.

## F.2   One-sided recommendation: Lastfm-15k dataset

We replicate the experiments on a larger dataset to verify our conclusions at a larger scale. We consider another Lastfm dataset from Celma [11], which includes $360k$ users and $180k$ items (artists). We select the top $15,000$ users and items having the most interactions, so we refer to this dataset as Lastfm-15k. We apply exactly the same experimental protocol as for Lastfm-2k, with the same range of hyperparameters for the different methods.

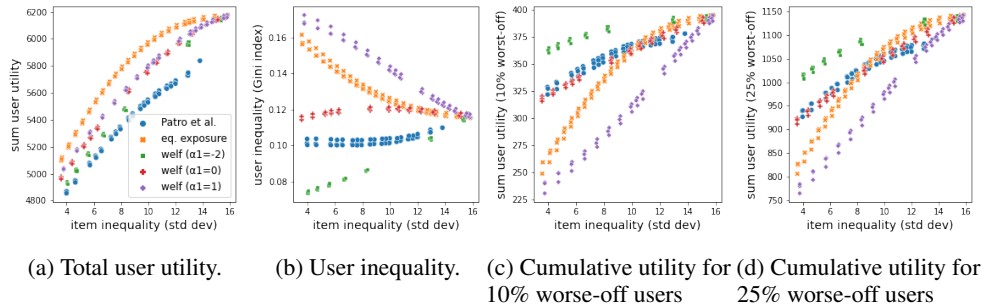

(a) Total user utility.    (b) User inequality.    (c) Cumulative utility for 10% worse-off users    (d) Cumulative utility for 25% worse-off users

Figure 8: Focus on user fairness on Lastfm-2k, measuring item inequality with standard deviation rather than Gini index. We observe a similar relative behavior between *welf* and *Patro et al.*, but now equality of exposure is optimal on the total utility/item inequality trade-off since it corresponds exactly to the objective of the algorithm. Nonetheless, *welf* $\alpha_1 = -2$ still obtains higher performance on 10%-25% worse-off users, showing that *welf* offers a larger range of trade-offs than *eq. exposure*.

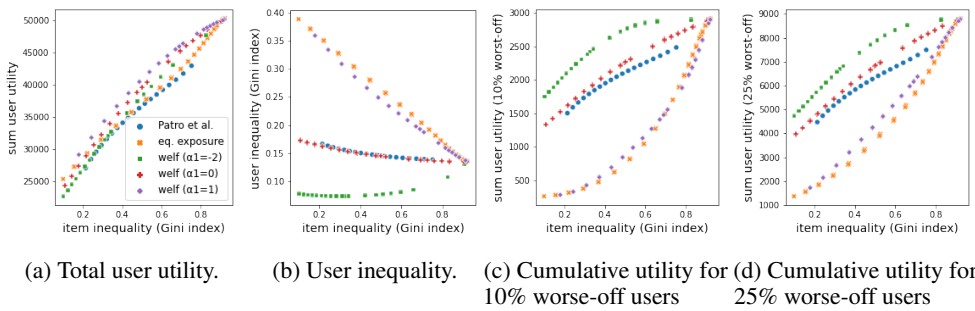

(a) Total user utility.    (b) User inequality.    (c) Cumulative utility for 10% worse-off users    (d) Cumulative utility for 25% worse-off users

Figure 9: Results on Lastfm-15k when measuring the inequality between items with the Gini coefficient.

**Results** Fig. 11 and 10 show the results obtained by *welf*, *Patro et al.* and *eq. exposure*. The conclusions are similar to those on Lastfm-2k, with the results of *welf* $\alpha = 0$ being more uniformly better than those of *Patro et al.*, even though overall similar. *welf* $\alpha_1 = -2$ dominates in terms of user utility on worse-off users. *welf* and *eq. exposure* still find different trade-offs, with *welf* dominating *eq. exposure* when inequality between items is measured by the Gini index, and *eq. exposure* dominating *welf* when inequality is measured by the standard deviation.

### F.3 One-sided recommendation: Movielens dataset

We provide additional results on the **MovieLens**-20m dataset [24], which contains ratings on a 5-star scale of movies by real users. To simulate a collaborative filtering task with implicit feedback similar

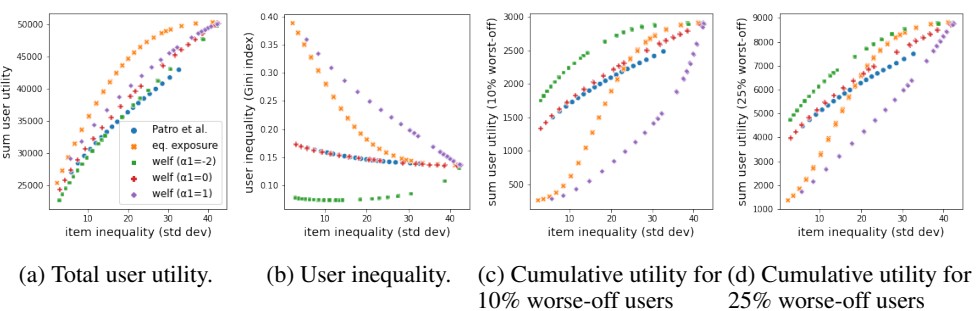

(a) Total user utility.    (b) User inequality.    (c) Cumulative utility for 10% worse-off users    (d) Cumulative utility for 25% worse-off users

Figure 10: Results on Lastfm-15k when measuring inequalities between items with the standard deviation.

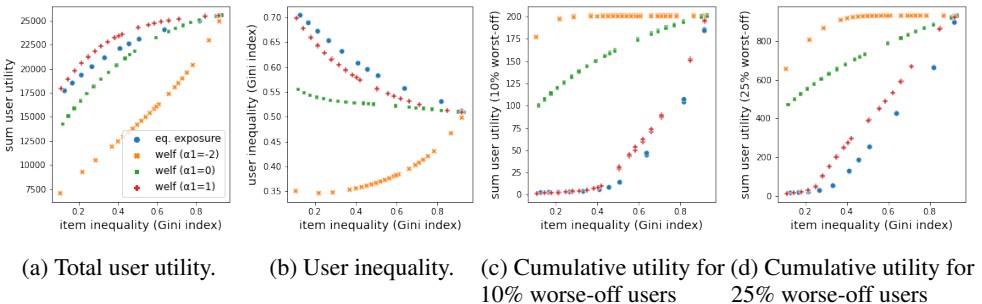

(a) Total user utility.  (b) User inequality.  (c) Cumulative utility for 10% worse-off users  (d) Cumulative utility for 25% worse-off users

Figure 11: Results on Movielens when measuring the inequality between items with the Gini coefficient.

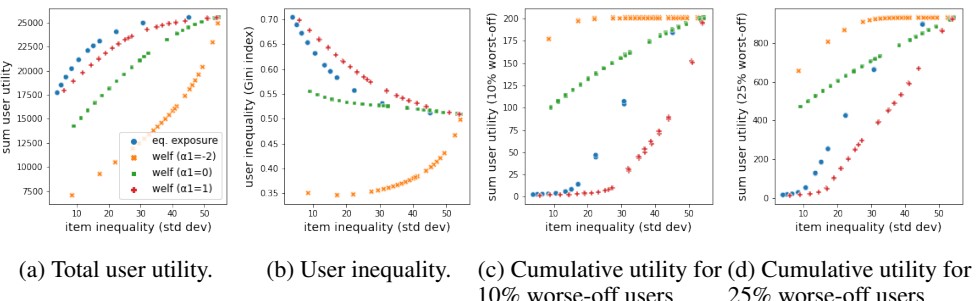

(a) Total user utility.  (b) User inequality.  (c) Cumulative utility for 10% worse-off users  (d) Cumulative utility for 25% worse-off users

Figure 12: Results on Movielens when measuring inequalities between items with the standard deviation.

to Last.fm, we consider missing ratings as negative feedback and the task is to predict positive values. Since ratings $< 3$ are usually considered as negative [38, 61], we set ratings $< 3$ to zero, resulting in a dataset with preference values among $\{0, 3, 3.5, 4, 4.5, 5\}$. As for Lastfm-15k, we select the top $15,000$ users and items with the most interactions. For the inference and evaluation of rankings, we follow the same protocols as for Last.fm.

The experimental protocol is the same as for Lastfm-2k and Lastfm-15k except that we do not run the algorithm by [47] because its runtime was prohibitive.

**results** The results are qualitatviely similar to those on Lastfm-2k and Lastfm-15k except that the trends are magnified. *welf* $\alpha = 1$ and *eq. exposure* seem more similar, with *welf* $\alpha = 0$ dominating the trade-off total utility/item iniequality when item inequality is measured with the Gini index, and *eq. exposure* dominating this trade-off when item inequality is measured with standard deviation. *welf* $\alpha = -2$ has great performance on worse-off users compared to *eq. exposure* or *welf* with larger $\alpha$, but also comes at a significant cost in terms of total user utility, which is very rapidly driven down.

### F.4 Reciprocal recommendation: Twitter-13k dataset

We now provide the full details of the experiments on Twitter presented in Section 5.2 of the main body. Given the lack of common benchmark for reciprocal recommendation [45], we generate a reciprocal recommendation task for people-to-people recommendation problems based on the social network Twitter. We use the Higgs Twitter-13k dataset which includes (directed) follower relationships between users.[18] We keep users having at least 20 mutual follows, resulting in a subset of 13k users. We use the directed links to estimate the probability $\phi_{ij}$ that $i$ follows $j$, and the (symmetric) probability of a mutual follow, which is $\mu_{ij} = \phi_{ij} \times \phi_{ji}$. As in the experiments for one-sided recommendation, we split the dataset into train/validation/test sets, each including $70\%/10\%/20\%$ of the *directed* follower links. We create three random uniform splits, corresponding to three different seeds.

---

[18]It was collected following the discovery of the Higgs boson in July, 2012.

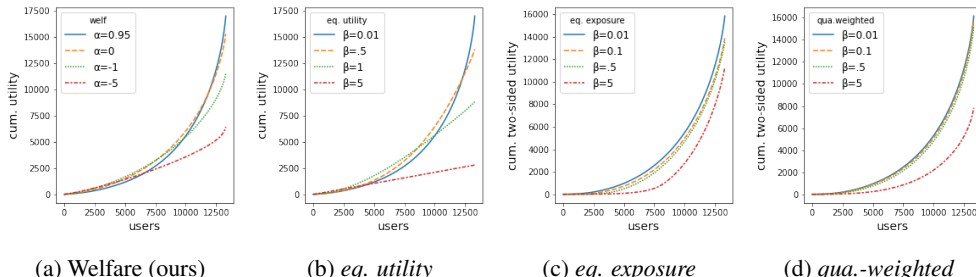

|  (a) Welfare (ours) | (b) *eq. utility* | (c) *eq. exposure* | (d) *qua.-weighted* |

Figure 13: representative trade-offs achieved by the various compared methods on Twitter-13k. Exposure-based approaches (*qua.-weighted* and *eq. exposure*) do not yield interesting trade-offs as they are unable to increase the utility of worse-off users. The trade-offs achieved by the *welf* and *eq. utility* are different. Equal utility rapidly generates near-flat curves without really focusing on the very first users, while *welf* increases the utility of the worst-off users while keeping the total utility relatively high.

Estimates $\hat{\phi}_{ij}$ are built with logistic matrix factorization[19] [32] trained on the train set with hyperparameter selection on the validation set. The number of latent factors is chosen in $[16, 32, 64, 128]$, the regularization in $[0.1, 1., 10., 20., 50.]$, and the confidence weighting parameter in $[0.1, 1., 10., 100.]$. Rankings are inferred from all estimated mutual preferences $\hat{\mu}_{ij} = \max(\hat{\phi}_{ij}\hat{\phi}_{ji}, 0)$. For each ranking method, the Frank-Wolfe algorithm is run with $5000$ iterations, and the number of recommendation slots is set to $40$. As for one-sided recommendation, rankings are estimated on estimated mutual preferences taken as ground truth.

We generate different trade-offs with *welf* by varying $\alpha$ in $[0.99, 0.9, 0.75, 0.5, 0.25, 0, -0.25, -0.5, -0.6, -0.7, -0.8, -0.9, -1.0]$, $[-1.1, -1.25, -1.5, -1.75, -2.0, -2.5, -3, -5, -10, -15, -16, -17, -18]$. For all other methods, we vary $\beta$ in $[0.01, 0.1, 0.2, 0.3, 0.4, 0.5, 0.6, 0.7, 0.8, 0.9, 1, 1.1, 1.25, 1.5, 2, 5, 10, 50, 100]$.

All presented results are obtained by averaging performance over the three seeds.

**Results**  Figure 13 presents the trade-offs achieved by the different methods on Twitter-13k. As expected, *qua.-weighted* and *eq. exposure* do not exhibit a good behavior: stronger penalties lead to more dominated curves where the utility of every user is decreased. This is because constraining item exposure is not meaningful in reciprocal recommendation, where the relevant utility is the two-sided utility. The trade-offs achieved by the *welf* and *eq. utility* are different. Equal utility rapidly generates near-flat curves without really focusing on the very first users, while *welf* increases the utility of the worst-off users while keeping the total utility relatively high.

### F.5  Reciprocal recommendation: Epinions dataset

We present additional experiments on reciprocal recommendation with the Epinions dataset [49]. Epinions.com is a consumer review site with a who-trust-whom network, and the dataset gathers (directed) trust relationships between members of the platform. Here, we consider the task of finding mutual trust links. We keep users having at least 20 mutual trust links, resulting in a subset of 800 entities. For the inference and evaluation of rankings, we use the same protocols as for the Twitter experiments described in the previous subsection. The experimental parameters are the same as for the Twitter-13k dataset.

**Results**  Figure 14 presents the trade-offs achieved by the different methods on Epinions. As expected, *qua.-weighted* and *eq. exposure* do not exhibit a good behavior: stronger penalties lead to more dominated curves where the utility of every user is decreased. In Figure 15 plots the equivalent of Fig. 4. The results are similar: all of *qua.-weighted*, *eq. exposure* and *eq. utility* have dominated curves. We also observe that in the more interesting region where we are closer to the maximum achievable utility, *welf* optimizes better the utility of worse-off users. Yet, in that region, there is no strict dominance of *welf* over *eq. utility*.

---

[19]Using the Python library Implicit (MIT License).

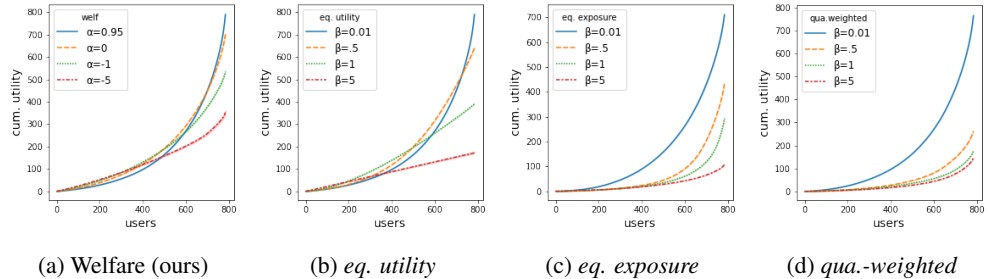

(a) Welfare (ours)  (b) *eq. utility*  (c) *eq. exposure*  (d) *qua.-weighted*

Figure 14: representative trade-offs achieved by the various compared methods on Epinions. The results are qualitatively similar to those on Twitter-13k.

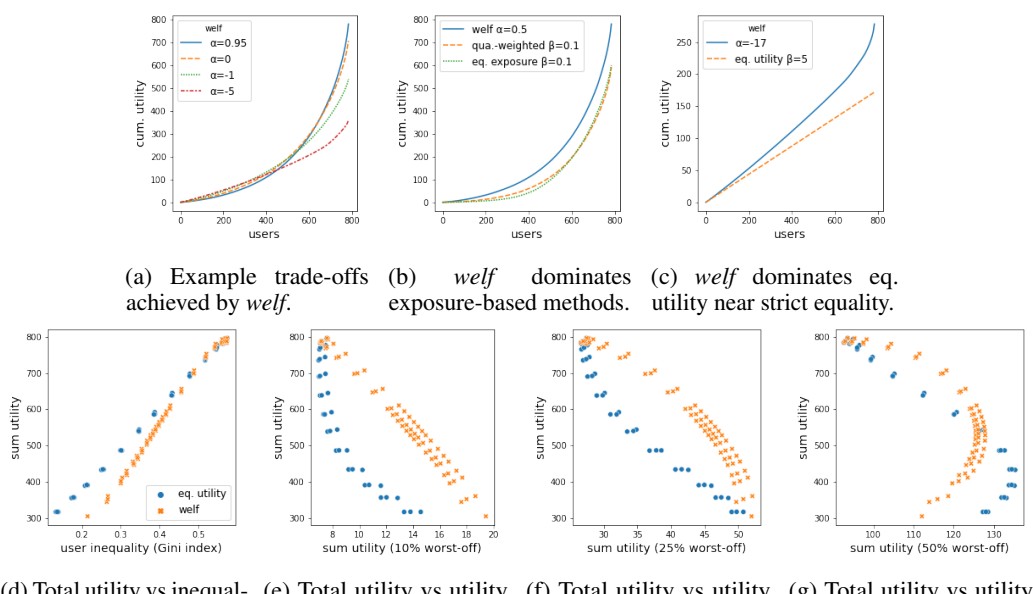

(a) Example trade-offs achieved by *welf*.

(b) *welf* dominates exposure-based methods.

(c) *welf* dominates eq. utility near strict equality.

(d) Total utility vs inequality.

(e) Total utility vs utility of 10% worse-off users.

(f) Total utility vs utility of 25% worse-off users.

(g) Total utility vs utility of 50% worse-off users.

Figure 15: Results on the epinions dataset.

## G  Pairwise vs pointwise penalties

Our penalty-based approach uses the penalty $\sqrt{D(\boldsymbol{u})}$ with:

$$D(\boldsymbol{u}) = \sum_{j \in \mathcal{I}} \left( u_j - \frac{1}{|\mathcal{I}|} \sum_{j' \in \mathcal{I}} u_{j'} \right)^2.$$

Some authors use $D'(\boldsymbol{u}) = \sum_{(j,j') \in \mathcal{I}^2} |u_j - u'_j|$ instead of $\sqrt{D(\boldsymbol{u})}$ [55, 42, 6], but it is less computationally efficient than our penalty because it involves a quadratic number of terms.

The penalties are similar in that they are related to well-known measures of inequalities:

- $\frac{D'(\boldsymbol{u})}{2|\mathcal{I}| \sum_{j \in \mathcal{I}} u_j}$ is the Gini index of $\boldsymbol{u}_{\mathcal{I}}$ [19], which, up to an affine transform is the area under the Lorenz curve.

- $D(\boldsymbol{u})$, which is (up to a constant) the variance of $\boldsymbol{u}_{\mathcal{I}}$ is part of the family of additively decomposable inequality measures [52]. We use $\sqrt{D(\boldsymbol{u})}$ to scale the penalty with the sum of users' utilities.

Note that $\sqrt{D(\boldsymbol{u})}$ and $D'(\boldsymbol{u})$ have the same dependency to the overall scale of the utilities (i.e., multiplying all utilities by a constant factor has the effect of multiplying both penalties by the same

factor). Since both penalties drive towards equality, it is straightforward to show that the results of Section 3 as $\beta \to \infty$ also apply to $D'(\boldsymbol{u})$.

## H   Exposure constraints at the level of every ranking

The notions of fairness of exposure are sometimes defined with item-side constraints defined at the level of *every ranking* [54, 6]. We give here the examples of constraints for equality of exposure and quality-weighted exposure:

$$\underset{\text{exposure}}{\text{equality of}} \qquad P^{\text{expo}} \in \underset{P \in \mathcal{P}}{\operatorname{argmax}} \sum_{i \in \mathcal{N}} u_i(P) \qquad \text{u.c. } \forall (i,j) \in \mathcal{N} \times \mathcal{I}, P_{ij}v = \frac{\|v\|_1}{|\mathcal{I}|}$$

$$\underset{\text{exposure}}{\text{quality-weighted}} \qquad P^{\text{qua}} \in \underset{P \in \mathcal{P}}{\operatorname{argmax}} \sum_{i \in \mathcal{N}} u_i(P) \qquad \text{u.c. } \forall (i,j) \in \mathcal{N} \times \mathcal{I}, P_{ij}v = \frac{\mu_{ij} \|v\|_1}{\sum_{j' \in \mathcal{I}} \mu_{ij'}}$$

The advantage of this formulation is that it leads to optimization problems that can be solved locally for every user, since there is no dependency between user rankings through item utility anymore.

However, applying the exposure criterion at the level of every ranking effectively applies a different notion of fairness. In our setting, this corresponds to defining a separate recommendation task for every user, i.e., taking $|\mathcal{N}| = 1$. The welfare function then mediates, within a single ranking, between the user utility and the utility of the different items.

When evaluated on exposures aggregated over all users, as we do in the paper, applying the fairness constraints at the level of individual rankings can lead to drastic reductions of user utility for no benefit in terms of total item exposure. This is summarized in the following result, which shows that there exists problems for which the optimal rankings for every $\theta \in \Theta$ satisfy the constraints of equality of exposure and quality-weighted exposure as we define them in Section 3, but when applying the constraints at the level of every ranking, it has the effect of reducing user utility. In the proposition, we use the notation of the objective function for parity of exposure $F_\beta$ and $F_\beta^{\text{qua}}$ of Section 3.

**Proposition 9.** *For every $d \in \mathbb{N}_*$ and every $N \in \mathbb{N}_*$, there is a one-sided recommendation task with $d+1$ items and $N(d+1)$ users such that, $\forall \theta \in \Theta$:*

$$\forall \boldsymbol{u}^\theta \in \operatorname{argmax}_{\boldsymbol{u} \in \mathcal{U}} W_\theta(\boldsymbol{u}), \forall \beta > 0 \text{ we have: } \boldsymbol{u}^\theta \in \underset{\boldsymbol{u} \in \mathcal{U}}{\operatorname{argmax}} F_\beta(\boldsymbol{u}) \text{ and } \boldsymbol{u}^\theta \in \underset{\boldsymbol{u} \in \mathcal{U}}{\operatorname{argmax}} F_\beta^{\text{qua}}(\boldsymbol{u}),$$

*and*

$$\sum_{i \in \mathcal{N}} u_i(P^{\text{expo}}) = \frac{2}{d+1} \sum_{i \in \mathcal{N}} u_i^\theta \qquad and \qquad \sum_{i \in \mathcal{N}} u_i(P^{\text{qua}}) = (\frac{1}{2} + \frac{1}{d}) \sum_{i \in \mathcal{N}} u_i^\theta.$$

In other words, applying the constraints at the level of every ranking might lead to a drastic decrease of user utilities, even in tasks where satisfying the constraints on average over users (as we do in this paper) does not conflict with the optimal ranking.

*Proof.* We describe the problem with $N = 1$, the general case is obtained by repeating the preference pattern. Let us consider a task with $d+1$ users, $d+1$ items and a single recommendation slot. Let $i_1, \ldots, i_{d+1}$ be the user indexes, and $j_1, \ldots, j_{d+1}$ the item indexes. The preferences are defined as:

$$\forall k \in [\![d+1]\!], \mu_{i_k j_k} = 1 \qquad \qquad \forall j \neq j_k, \mu_{i_k j} = \frac{1}{d}.$$

All items have the same quality. For every $\theta \in \Theta$, $\boldsymbol{u}^\theta$ is given by the utilitarian ranking, which gives probability 1 to item $j_k$ for user $i_k$, which leads to optimal user utility $u_i^\theta = 1$ and equal exposure to every item $u_j^\theta = 1$. Since the quality is the same for all items (equal to $1 + d\frac{1}{d}$), the ranking for $\boldsymbol{u}^\theta$ satisfies both equality of exposure and quality-weighted exposure constraints. Thus, for every $\beta > 0$, $\boldsymbol{u}^\theta \in \operatorname{argmax}_{\boldsymbol{u} \in \mathcal{U}} F_\beta(\boldsymbol{u})$ and $\boldsymbol{u}^\theta \in \operatorname{argmax}_{\boldsymbol{u} \in \mathcal{U}} F_\beta^{\text{qua}}(\boldsymbol{u})$.

On the other hand, satisfying equality of exposure at the level of every ranking requires $P_{ij}^{\text{expo}} = \frac{1}{d+1}$ for every user $i$ and item $j$, which leads to $u_i(P^{\text{expo}}) = \frac{1}{d+1} + d \times \frac{1}{d} \times \frac{1}{d+1} = \frac{2}{d+1}$ for every user.

For quality-weighted exposure for every ranking, it leads to:

$$\forall k \in [\![d+1]\!], P^{\mathrm{qua}}_{i_k j_k} = \frac{1}{2} \qquad\qquad \forall j \neq j_k, P^{\mathrm{qua}}_{i_k j} = \frac{1}{d}$$

and thus a user utility $u_i(P^{\mathrm{qua}}) = \frac{1}{2} + d \times \frac{1}{d} \times \frac{1}{d} = \frac{1}{2} + \frac{1}{d}$. $\qquad\qquad\square$

Notice that in the examples of the proof, the global exposure of items is constant in $P^{\mathrm{expo}}$ and $P^{\mathrm{qua}}$, as well as in the ranking given by optimal welfare. So from the point of view of our definitions of utility, applying the constraints at the level of every ranking only decreased user utility for the benefit of no items. Yet, we re-iterate that applying item-side fairness at the level of every ranking might be meaningful in some contexts. The goal of this section is to highlight the difference between using global and local definitions of item utilities.