# OpenReview forum: "Two-sided fairness in rankings via Lorenz dominance"
_NeurIPS.cc/2021/Conference — NeurIPS 2021 Poster_

### Official Review · Reviewer_wTCs · 2021-07-16

**Rating:** 6
**Confidence:** 3

**Summary:**

The authors study fairness in recommendation lists, where fairness is considered on both the part of consumers / users, as well as producers / items. They define fairness as a combination of Pareto efficiency and equity given that total level of utility, and note that non-dominated Lorenz curves satisfy their definition of fairness. They demonstrate that their method can be efficiently computed and show that it does well on real-world datasets.


**Ethical Concerns:**

None.

**Limitations And Societal Impact:**

Yes.

**Main Review:**

The authors study a natural consideration in matching problems where you must balance fairness on the part of both the users and advertisers simultaneously.

The definition of fairness introduced by the authors makes sense in a two-sided market, and they bring up a good parallel to Pareto efficiency. However, Pareto efficiency (and Lorenz domination) can be a relatively weak property in the sense that a Pareto or Lorenz curve with one small "good part" may not ever be dominated by any other curve even though those curves are better "overall".

I found the paper to be quite dense and notationally heavy, and some results were hard to interpret without the appendix.

Around line 103, the analogies explaining why \mu_{ij} = \mu_{ji} in two-sided preferences don't exactly fit (at least not to me). Prospective employers and prospective employees derive different utilities from an interview; in general, people in relationships can derive different utilities as well.

Around line 160, could you please explain the alphas a bit better? It seems like each alpha should be between 0 and 1 in order to get a strictly concave function, and the experiments show a range of alphas -- how did you choose them in asymmetric settings?

I also found the figures in the experimental section a bit hard to interpret. Would it be helpful / informative to plot "perfectly equitable" Lorenz curves where applicable to more clearly illustrate domination?


Minor comments:

44 and 46: either "can lead to a decrease in the utility" or "can decrease the utility"
47: worst-off
89: users
91: cardinality
92: the (i,j)-th
97: either? stress that this "either" is based on previous lines of work; it'll be clear in this paper depending on context
103: don't understand the analogies. different utilities to each side...suggest something like
201: of the most satisfied users
224: qualitatively

**Time Spent Reviewing:**

2

---

> ### Author Response · Authors · 2021-08-06
> **Author Response to Reviewer wTCs**
>
> Thank you for your valuable feedback.
>
> 1/ “However, Pareto efficiency (and Lorenz domination) can be a relatively weak property in the sense that a Pareto or Lorenz curve with one small "good part" may not ever be dominated by any other curve even though those curves are better "overall".”
>
> The reviewer points to the property that both Pareto and Lorenz dominance are only partial orders -- in particular, they do not specify how to choose between utility profiles with Lorenz curves that intersect. Thus, in practice we need finer grained evaluations of the trade-offs to make the final decision. Our approach does resolve the issue pointed out by the reviewer: 1) the fairness criterion is that we should only accept non-dominated Lorenz curves, 2) the practical algorithm uses a welfare function, the parameters of which controls which utility profile we want to choose among the non-dominated ones. Thus, what the reviewer calls "better overall" is controlled by the choices of $\alpha_1, \alpha_2$ and $\lambda$.
>
> 2/ “why \mu_{ij} = \mu_{ji} in two-sided preferences”
>
> Please, note that our results do not rely on this assumption. We discuss this special case because it is a common assumption in the literature on reciprocal recommendation (see e.g., the survey [39]).
>
> 3/ “It seems like each alpha should be between 0 and 1 in order to get a strictly concave function”
>
> The range of alpha is $-\infty < \alpha < 1$. For $\alpha < 0$, the welfare function is in $-x^\alpha$ (L160), which is strictly concave.
>
> 4/ “how did you choose [the alphas] in asymmetric settings?”
>
> We chose $\alpha_1 = \alpha_2$ in our experiments (L309), because we obtained satisfying trade-offs with the welfare approach while limiting the number of parameters to simplify the analysis. In theory though, considering distinct values $\alpha_1, \alpha_2$ for users and items is interesting because it leads to item-side leximin-optimal solutions (L180 and App C.2 Prop. 7).
>
> 5/ “I also found the figures in the experimental section a bit hard to interpret.”
>
> A ranking is deemed unfair by the Lorenz efficiency criterion if (a) in one-sided recommendation, both its user-side and item-side Lorenz curves are dominated, (b) in reciprocal recommendation, its two-sided utility Lorenz curve is dominated. In Figure 2 (right column), we thus observe that quality-weighted exposure fails the criterion since *both* user and item-side curves are dominated by the welfare approach. In Figure 2,3 (left column), we show that the parameters of the welfare function guide the choice between intersecting curves: either the curve with “one small good part” or which appear “better overall”. By Proposition 1, the solutions obtained with the welfare approach for any choice of parameters are fair. These trade-offs can also be observed on the figures in Appendix F.

---

> > ### Author Response · Authors · 2021-08-24
> > **Did our response address your concerns?**
> >
> > Dear Reviewer wTCs,
> >
> > We would be grateful if you could confirm whether our response has addressed your concerns, and let us know if any issues remain. To recap our response, we:
> > * provided guidance on the interpretation of Lorenz efficiency and Lorenz curves
> > * explained the range and choice of $\alpha$.

---

> > > ### Comment · Reviewer_wTCs · 2021-08-26
> > > **Response**
> > >
> > > Apologies for the late reply! Your very detailed response has addressed some of my concerns. I would have appreciated slightly more exposition around both the experimental results and the choice of alphas in the papers but the response helps a lot. Thanks!

---

### Official Review · Reviewer_WERu · 2021-07-18

**Rating:** 6
**Confidence:** 4

**Summary:**

This paper proposes a new ranking framework that concerns the fairness of all parties (i.e., users and items) participating in recommender systems. Inspired by techniques used in welfare economics, the authors formulate a convex optimization that yields Pareto-efficient and equitable (within each parties) utilities. To demonstrate the superiority over existing methods, they consider some special cases where the proposed approach is shown to achieve better utilities than the baseline methods. They provide an efficient implementation based on the Frank-Wolfe algorithm, together with theoretical results on tractability and convergence.

**Limitations And Societal Impact:**

The paper discussed limitations and societal impacts adequetly in conclusion.

**Main Review:**

Strengths:
S1: The main idea (of employing Lorenz dominance to implement fair utility) is insightful.
S2: The proposed formulation is elegant: it covers both one-sided and reciprocal recommendation systems.
S3: Theoretical guarantees of the considered algorithm are provided in Theorems 1 and 2.

Weaknesses:
W1: Improvement over the considered baselines (particularly for equal_expo and equal_util) seems marginal. This blurs the superiority gained in Section 4 and hence weaken the motivation of using the proposed framework instead of a more simple item-side fairness scheme.
W2: Comparison with prior two-sided fairness algorithms (such as [41]) are missing.
W3: "Tradeoffs" are unclear in Figure 2 and Figure 3. As described in Section 6, now the left column in Fig. 2 exhibits tradeoff between user and item utilities depending on alpha, but I am not sure this tension still remains when "alpha_1" and "alpha_2" take different values. It seems that if we take "alpha_1 = 0.95" and "alpha_2 = 0", then both user and item utilities yield the best curves with no trade-off. Figure 3 only shows aggregated two-sided utilities so tension between two utilities is impossible to see.

The technical content of this paper appears to be correct.
The paper was overall well-written yet would benefit from more proofreading. More precisely:
- Please remove "is” from "is is the value” in line 96 of page 3.


**Time Spent Reviewing:**

6

---

> ### Author Response · Authors · 2021-08-06
> **Author Response to Reviewer WERu**
>
> Thank you for your valuable comments.
>
> W1: As stated in the introduction (L43), our main claim is that the widely adopted criterion of quality-weighted exposure [53,45] (a.k.a. equity of attention [7] or merit-based fairness [46]) is inequitable because it can decrease the utility of every user *and* increase inequalities between items, both in theory and practice. Equal exposure (or equal item utility) [45] comparatively received little attention, likely because most papers focus on strict fairness constraints rather than trade-offs. It is a notable result of the paper that these criteria yield reasonable trade-offs, in contrast to quality-weighted exposure. The interest of our welfare approach lies in its stronger theoretical properties, which prevent the risk of inefficiency and inequity of previous approaches.
>
> The dominance relations may appear more clearly from the plots in linear scale available in Appendix F (Fig. 10-11 for reciprocal recommendation on Twitter). We chose to include plots in log scale in the main paper because they allow better visualization of dominance between curves for worse-off users.
>
> W2: [41] does not produce rankings. Further, [41] considers two distinct criteria for users and for items (respectively envy-freeness and max-min share) without interpersonal comparison of utilities, and thus does not seek to trade off utility between them. We are currently integrating the algorithm of [41] into our code and will post the results as soon as we have them.
>
> W3: In theory, we showed that $\alpha_1$ and $\alpha_2$ played different roles: more precisely, having $\alpha_1$ constant and $\alpha_2 \rightarrow \infty$ allows to reach leximin-optimal rankings (L180 and Prop. 7, App. C2). In preliminary experiments, we observed that good trade-offs could already be obtained with $\alpha_1=\alpha_2$, and we kept this configuration to limit the number of degrees of freedom to analyze.

---

> > ### Author Response · Authors · 2021-08-10
> > **Additional experiments: comparison with FairRec [41]**
> >
> > We integrated the FairRec algorithm from [41] to our code and ran it on Lastfm-2k and Epinions -- the algorithm takes too much time on larger datasets such as Twitter-13k (at least 8 days for $K=400$). Since FairRec does not produce rankings, we took the recommendation list with the given order.
> >
> > For one-sided recommendation on Lastfm, by varying the max-min share parameter of FairRec, we observe that FairRec produces trade-offs akin to equal exposure and the welfare approach, because the rankings inferred by the latter methods are likely envy-free.
> >
> > For reciprocal recommendation on Epinions, we observe that FairRec is clearly dominated. We summarize our results in the table below. For a similar amount of two-sided utility, the welfare approach distributes utility more equitably than FairRec, as indicated by the area under the Lorenz curve.
> >
> >
> > |                                      | FairRec ($\alpha=0.7$) [41] | Welfare ($\alpha=0.95$) |
> > |--------------------------------------|-----------------------------|-------------------------|
> > | Total two-sided utility              | $1.392\times10^6$           | $1.393\times10^6$       |
> > | Area Under Curve (two-sided utility) | $345\times10^6$             | $380\times10^6$         |

---

> > > ### Author Response · Authors · 2021-08-24
> > > **Did our response address your concerns?**
> > >
> > > Dear Reviewer WERu,
> > >
> > > We would be grateful if you could confirm whether our response has addressed your concerns, and let us know if any issues remain. To recap, we responded to W1 and W3, and we addressed W2 by providing additional experiments with the algorithm of paper [41]. Thank you in advance.

---

> > > > ### Comment · Reviewer_WERu · 2021-08-26
> > > > **After authors' response**
> > > >
> > > > Thanks for your detailed answer to my comments. I found them clarifying, but still have a concern about experimental results. As pointed in W3 of my initial comment, I don’t understand what tradeoff one can see in reciprocal recommendation results. More precisely, L336 indicates that the two left plots in Figure 3 exhibits trade-off behaviors of two methods (welfare and equal_expo), but I could not find any such trade-off behaviors between user and item utilities, which are well represented in one-sided recommendation scenarios as in Figure 2. I would be more convinced if the authors address this point.

---

> > > > > ### Author Response · Authors · 2021-08-26
> > > > > **Response to Reviewer WERu on the trade-offs in reciprocal recommendation**
> > > > >
> > > > > In reciprocal recommendation, the relevant quantity is the two-sided utility (recall that in reciprocal recommendation users and items are the same). The trade-off is then how (two-sided) utility is distributed between users. In that case, decreasing $\alpha$ has the effect of increasing the utility of the least served users. (The mechanism is that least-served users are allocated more exposure, which increases their two-sided utility, at the expense of best-served users).
> > > > >
> > > > > The trade-offs that arise is that by decreasing $\alpha$, we reallocate utility from the best-served users to the least served users, making the Lorenz curve higher on the left but lower on the right. In other words, we have more equality at the expense of total (sum) utility.
> > > > >
> > > > > In figure 3 (left column), this trade-off is not very visible because the curves are close to each other for different values of alpha, it is slightly more visible in figure 11 in appendix F which shows the curves in linear scale.
> > > > >
> > > > > In order to offer a better illustration of the trade-offs, we computed the area under the Lorenz curve for different values of $\alpha$, as well as the sum of the two-sided utilities.
> > > > >
> > > > >
> > > > > | Welfare                              | $\alpha=0.95$     | $\alpha=0.7$      | $\alpha=0$        | $\alpha=-0.5$     |
> > > > > |--------------------------------------|-------------------|-------------------|-------------------|-------------------|
> > > > > | Total two-sided utility              | $2.727\times10^6$ | $2.707\times10^6$ | $2.672\times10^6$ | $2.656\times10^6$ |
> > > > > | Area Under Curve (two-sided utility) | $1.375\times10^9$ | $1.520\times10^9$ | $1.608\times10^9$ | $1.629\times10^9$ |
> > > > >
> > > > >
> > > > > We thus see that increasing equity (redistribution) comes at the expense of total utility (hence a trade-off that we need to choose from in practice). The good news is that we can significantly increase the area under the Lorenz curve at little cost in terms of sum-utility.
> > > > >
> > > > >
> > > > > All the trade-offs described above are “fair”. However when we compare our welfare approach to quality-weighted exposure, we see that welfare dominates quality-weighted both in terms of area under the Lorenz curve and total utility — which makes the trade-offs of quality-weighted exposure unfair according to our criteria, because some users have lost a lot of utility for suboptimal redistribution.
> > > > >
> > > > > |                                      | qual, $\beta=0.2$ | qual, $\beta=0.5$ | welf, $\alpha=0.7$    |
> > > > > |--------------------------------------|-------------------|-------------------|-----------------------|
> > > > > | Total two-sided utility              | $2.663\times10^6$ | $2.493\times10^6$ | **$2.707\times10^6$** |
> > > > > | Area Under Curve (two-sided utility) | $1.366\times10^9$ | $1.274\times10^9$ | **$1.520\times10^9$** |

---

> > > > > > ### Author Response · Authors · 2021-08-30
> > > > > > **Did our response clearly address your last comment?**
> > > > > >
> > > > > > Dear Reviewer WERu,
> > > > > >
> > > > > > Did our response clarify the trade-offs involved in reciprocal recommendation? Thank you in advance for letting us know if any issues remain.

---

### Official Review · Reviewer_oh3N · 2021-07-19

**Rating:** 6
**Confidence:** 4

**Summary:**

The authors of this work study fairness in two-sided platforms that utilize ranking as a mechanism for improve experience and utility of users and “items” (that can also be users). They introduce a new framework using the notion of Lorenz dominance through equity and efficiency. The work also contributes a novel algorithm (based on Frank-Wolfe) and show experimental evidence for its performance.

**Main Review:**

Originality: To the best of my knowledge, the results & contributions of the study are novel.

Quality: The work has a notable amount of contribution and all of them are supported by theoretical results (with a lot of detailed proofs in Supp. Mat.). Also, the authors provide experimental evaluation.

Clarity: On the one hand, the paper is well structured, with a good story line and introduction of all required theoretical definitions, theorems and experimental setup. On the other hand, the authors permanently consider several variants of their setup "one-sided recommendation", "reciprocal recommendation”, "one-sided preferences”, and "two-sided preferences"

For instance, Line 159 & Line 161: “In our case where there are users and items …” and “In reciprocal recommendation”. This is one among a lot of other lines of examples. Sometimes, it leads to misunderstanding of what is the considered case in this work.

Is it possible to make the work more focused on one single particular setup, while other ones localize in a dedicated place of the paper (e.g., move to discussion)?

[CLARIFIED DURING REBUTTAL PERIOD] Line 283: It is not clear about “O(K) additional bytes for each iteration” (the proof in Appendix did not help). Even \tilda P is new for each iteration t, but all previous \tilda P (from steps t-1, t-2, t-3, etc) are not required to keep in memory (Am I right?). So, we can use the same memory for each iteration.


[CLARIFIED DURING REBUTTAL PERIOD] Significance: Overall, I appreciate the contribution of the work. However, I believe that it may significantly benefit for practice if it (at least) discusses how the results can be used in paid scenarios. Many ranking (recomm) systems are paid. So, how are these results related to such systems that are quite popular? Usually, every large system needs a monetization implying sponsored ranking / recommendations / suggestions / etc.: e.g., sponsored search, social networks, ecom, crowdsourcing platforms,…

Minor comments:
-	Line 130: I did not get “true” applied to “user”


**Time Spent Reviewing:**

8

---

> ### Author Response · Authors · 2021-08-06
> **Author Response to Reviewer oh3N**
>
> Thank you for your valuable feedback.
>
> 1/ “L283: O(K) additional bytes for each iteration”
>
> At each iteration, we store $\tilde{P}$ in memory to obtain a decomposition of the ranking tensor $P$ as a weighted sum of permutation matrices (L277). We keep this representation because it corresponds to the sparse representation of the solution: we represent a full stochastic ranking as a convex combination of Dirac over deterministic rankings, and we add one element to this representation at each iteration. Indeed, a consequence of Theorem 2 is that $\tilde{P}$ is a tensor with one permutation matrix for each user, which can be represented as one deterministic ranking per user (which is $O(K)$ for top-K rankings).
>
> 2/ “I believe that it may significantly benefit for practice if it (at least) discusses how the results can be used in paid scenarios.”
>
> Thank you for your suggestion. A setting that considers sponsored search auctions is indeed a relevant extension of our work. In our framework, we could see this as one-sided recommendation with two-sided preferences, meaning that only ads (items) are being recommended to users, but both advertisers (item producers) and users have meaningful preferences. Bids can be used as surrogate for item-side utilities, and then the goal is to trade off between user utility and advertiser bids. However, more elaboration is needed to properly define auction rules. We will add a discussion on this point, but a full analysis is left for future work.

---

> > ### Comment · Reviewer_oh3N · 2021-08-25
> > **Thank you for your clarification**
> >
> > Thank you for your clarification. I got both points. Considering the 2nd one, yes, it would be great to see this discussion in the text.

---

> > > ### Author Response · Authors · 2021-08-30
> > > **We are willing to address any pending issues**
> > >
> > > Dear Reviewer oh3N,
> > >
> > > Thank you for your comment, we will add point 2 to the discussion in the paper. If any issues remain, please let us know.

---

### Decision · Program_Chairs · 2021-09-27

**Decision:**

Accept (Poster)

**Comment:**

Reviewers were very appreciative of the melding of traditional economic concepts such as Lorenz dominance with the topical problem of two-sided fairness in ranking/matching platforms.  Parallels to Pareto efficiency and other realistic desiderata and concerns were also appreciated.  Some comparisons to baselines/related work seemed marginal or weak.  In a subsequent or final version, I would ask that the authors do indeed make the changes promised during the post-rebuttal discussion (e.g., addition of discussion of sponsored search / other paid settings, all typos/method clarifications, newer results, and so on).  This is a case where the rebuttal and discussion directly and positively influenced the reviewers' (and this AC's) opinion.